# POMINO-GEMS: A Research Product for Tropospheric NO₂ Columns from Geostationary Environment Monitoring Spectrometer

Yuhang Zhang[1], Jintai Lin[1], Jhoon Kim[2], Hanlim Lee[3], Junsung Park[3], Hyunkee Hong[4], Michel Van Roozendael[5], Francois Hendrick[5], Ting Wang[6,7], Pucai Wang[6,7], Qin He[8], Kai Qin[8], Yongjoo Choi[9], Yugo Kanaya[10], Jin Xu[11], Pinhua Xie[7,11], Xin Tian[12], Sanbao Zhang[13], Shanshan Wang[13], Siyang Cheng[14], Xinghong Cheng[14], Jianzhong Ma[14], Thomas Wagner[15], Robert Spurr[16], Lulu Chen[17], Hao Kong[1], Mengyao Liu[18]

[1]Laboratory for Climate and Ocean-Atmosphere Studies, Department of Atmospheric and Oceanic Sciences, School of Physics, Peking University, Beijing, 100871, China

[2]Department of Atmospheric Sciences, Yonsei University, Seoul, South Korea

[3]Pukyong National University, Busan, South Korea

[4]National Institute of Environmental Research, Incheon, South Korea

[5]Belgian Institute for Space Aeronomy (BIRA-IASB), Brussels, Belgium

[6]CNRC & LAGEO, Institute of Atmospheric Physics, Chinese Academy of Sciences, Beijing, 100029, China

[7]University of Chinese Academy of Sciences, Beijing, 100049, China

[8]School of Environment and Geoinformatics, China University of Mining and Technology, Xuzhou, Jiangsu, 221116, China

[9]Department of Environmental Science, Hankuk University of Foreign Studies, Yongin, South Korea

[10]Research Institute for Global Change, Japan Agency for Marine-Earth Science and Technology (JAMSTEC), Yokohama, 2360001, Japan

[11]Key Laboratory of Environmental Optics and Technology, Anhui Institute of Optics and Fine Mechanics, Chinese Academy of Science, Hefei, 230031, China

[12]Information Materials and Intelligent Sensing Laboratory of Anhui Province, Institutes of Physical Science and Information Technology, Anhui University, Hefei, Anhui, 230601, China

[13]Shanghai Key Laboratory of Atmospheric Particle Pollution and Prevention (LAP3), Department of Environmental Science and Engineering, Fudan University, Shanghai, 200433, China

[14]State Key Laboratory of Severe Weather & Institute of Tibetan Plateau Meteorology, Chinese Academy of Meteorological Sciences, Beijing, 100081, China

[15]Max Planck Institute for Chemistry, 55020, Mainz, Germany

[16]RT Solutions Inc., Cambridge, Massachusetts, 02138, USA

[17]College of Urban and Environmental Sciences, Peking University, Beijing, 100871, China

[18]R&D Satellite Observations Department, Royal Netherlands Meteorological Institute, De Bilt, the Netherlands

*Correspondence to*: Jintai Lin (linjt@pku.edu.cn)

## Abstract

Tropospheric vertical column densities (VCDs) of nitrogen dioxide (NO₂) retrieved from sun-synchronous satellite instruments have provided abundant NO₂ data for environmental studies, but such

data are limited by retrieval uncertainties and insufficient temporal sampling (e.g., once a day). The
Geostationary Environment Monitoring Spectrometer (GEMS) launched in February 2020 monitors $NO_2$
at an unprecedented hourly resolution during the daytime. Here we present a research product for
tropospheric $NO_2$ VCDs, referred to as POMINO-GEMS. We develop a hybrid retrieval method
combining GEMS, TROPOMI and GEOS-CF data to generate hourly tropospheric $NO_2$ slant column
densities (SCDs). We then derive tropospheric $NO_2$ air mass factors (AMFs) with explicit corrections for
surface reflectance anisotropy and aerosol optical effects, through parallelized pixel-by-pixel radiative
transfer calculations. Prerequisite cloud parameters are retrieved with the $O_2$-$O_2$ algorithm by using
ancillary parameters consistent with those used in $NO_2$ AMF calculations.
Initial retrieval of POMINO-GEMS tropospheric $NO_2$ VCDs for June–August 2021 exhibits strong
hotspot signals over megacities and distinctive diurnal variations over polluted and clean areas.
POMINO-GEMS $NO_2$ VCDs agree with the POMINO-TROPOMI v1.2.2 product ($R = 0.98$, and NMB
$= 4.9\%$) over East Asia, with slight differences associated with satellite viewing geometries and cloud
and aerosol properties affecting the $NO_2$ retrieval. POMINO-GEMS also shows good agreement with
OMNO2 v4 ($R = 0.87$, and NMB $= -16.8\%$) and GOME-2 GDP 4.8 ($R = 0.83$, and NMB $= -1.5\%$) $NO_2$
products. POMINO-GEMS shows small biases against ground-based MAX-DOAS $NO_2$ VCD data at
nine sites (NMB $= –11.1\%$) with modest or high correlation in diurnal variation at six urban and suburban
sites ($R$ from 0.60 to 0.96). The spatiotemporal variation of POMINO-GEMS correlates well with
mobile-car MAX-DOAS measurements in the Three Rivers' Source region on the Tibetan Plateau ($R =$
0.81). Surface $NO_2$ concentrations estimated from POMINO-GEMS VCDs are consistent with
measurements from the Ministry of Ecology and Environment of China for spatiotemporal variation ($R$
$= 0.78$, and NMB $= –26.3\%$) as well as diurnal variation at all, urban, suburban and rural sites ($R \geq 0.96$).
POMINO-GEMS data will be made freely available for users to study the spatiotemporal variations,
sources and impacts of $NO_2$.
**1. Introduction**
Tropospheric nitrogen dioxide ($NO_2$) is an important air pollutant. It threats human health, and
contributes to the formation of tropospheric ozone ($O_3$) and nitrate aerosols (Crutzen, 1970; Shindell et
al., 2009; Hoek et al., 2013; Chen et al., 2022). Satellite instruments provide observations of tropospheric
$NO_2$ on a global scale, and they have been extensively used to estimate emissions of nitrogen oxides
($NO_x$ = NO + $NO_2$) (Lin and Mcelroy, 2011; Beirle et al., 2011; Gu et al., 2014; Kong et al., 2022a),
surface $NO_2$ concentrations (Wei et al., 2022; Cooper et al., 2022), trends and variabilities (Richter et al.,
2005; Cui et al., 2016; Krotkov et al., 2016; Van Der A et al., 2017), and impacts on human health and
environment (Chen et al., 2021).

To date, most spaceborne instruments for $NO_2$ measurements, including the Global Ozone

Monitoring Instrument (GOME) (Burrows, 1999), the Ozone Monitoring Instrument (OMI) (Levelt et
al., 2006), the Global Ozone Monitoring Experiment 2 (GOME-2) (Callies et al., 2000) and the
TROPOspheric Monitoring Instrument (TROPOMI) (Veefkind et al., 2012), are mounted on sun-
synchronous low Earth orbit (LEO) satellites. These instruments passively measure backscattered
radiance from the Earth's atmosphere, and measurements at each ground location are done 1–2 times a
day. The Geostationary Environment Monitoring Spectrometer (GEMS) on board the Geostationary
Korea Multi-Purpose Satellite-2B (GK-2B) was successfully launched in February 2020. The instrument
provides measurements of $NO_2$ and other pollutants in the daytime on an hourly basis (Kim et al., 2020).
It complements LEO satellite observations by providing a more comprehensive picture of the daytime
evolution of $NO_2$.

There are three successive stages in the retrieval of tropospheric $NO_2$ vertical column densities

(VCDs) in the UV-Vis range based on satellite observations. The first step is to retrieve total $NO_2$ slant
column densities (SCDs) with spectral fitting techniques, such as the Differential Optical Absorption
Spectroscopy (DOAS). The SCD represents the abundance of $NO_2$ along the effective light path from
the sun through the atmosphere to the satellite instrument. Next, the contributions from stratospheric $NO_2$
to the total SCDs are removed in order to obtain tropospheric SCDs. Finally, the tropospheric SCDs are
converted to VCDs using calculated air mass factors (AMFs). The AMF calculations are highly sensitive
to the observation geometry, cloud parameters, aerosols, surface conditions and the shape of the $NO_2$
vertical distribution. Over polluted areas, errors in the retrieved tropospheric $NO_2$ VCDs are dominated
by the uncertainties in AMF calculations (Boersma et al., 2004; Lorente et al., 2016) associated with
aerosol optical effects, surface reflectance and a priori $NO_2$ vertical profiles (Zhou et al., 2010; Lin et al.,
2014; Lin et al., 2015; Vasilkov et al., 2016; Lorente et al., 2018; Liu et al., 2019; Liu et al., 2020;
Vasilkov et al., 2021).

The official GEMS retrieval algorithm for tropospheric $NO_2$ VCDs is developed by Lee et al. (2020).

The total $NO_2$ SCDs are retrieved using the DOAS technique. They are then converted to total $NO_2$ VCDs
by using a precomputed look-up table of box AMFs based on the linearized pseudo-spherical scalar and
vector discrete ordinate radiative transfer code (VLIDORT) version 2.6. Finally, stratosphere-
troposphere separation (STS) is performed to derive tropospheric $NO_2$. Validation results have shown the
overall capability of the official GEMS $NO_2$ algorithm (Kim et al., 2023), but several problems are also
reported, such as overestimation of total $NO_2$ SCDs and tropospheric $NO_2$ VCDs, and some degree of
striping in $NO_2$ retrieval data.
In this study, we present a research product which we name as POMINO-GEMS. This product is
built upon our Peking University OMI $NO_2$ (POMINO) algorithm which focuses on the tropospheric
AMF calculations and has been applied to OMI and TROPOMI (Lin et al., 2014; Lin et al., 2015; Liu et
al., 2019; Liu et al., 2020; Zhang et al., 2022). Here we extend the AMF calculation by constructing a
hybrid method to estimate tropospheric SCDs for GEMS. The hybrid method makes use of the total
SCDs from the official GEMS product, total SCDs and stratospheric VCDs from the official TROPOMI
product, as well as hourly stratospheric VCD data from the NASA Global Earth Observing System
Composition Forecast (GEOS-CF) v1 product. We validate our initial set of retrieval results for
tropospheric $NO_2$ VCDs in June-July-August (JJA) 2021, by using independent data of tropospheric $NO_2$
from the POMINO-TROPOMI v1.2.2, OMNO2 v4 and GOME-2 GDP 4.8 products, ground-based and
mobile-car MAX-DOAS measurements, and surface concentration observations from the Ministry of
Ecology and Environment (MEE) of China. We provide a simplified estimate of retrieval errors in the
end.
**2. Method and data**
**2.1 Construction of POMINO-GEMS retrieval algorithm**
Figure 1 shows the flow chart of POMINO-GEMS retrieval algorithm. There are two essential steps.
The first is to calculate tropospheric $NO_2$ SCDs on an hourly basis, through fusion of total SCDs from
the official GEMS v1.0 L2 $NO_2$ product, total SCDs and stratospheric VCDs from the TROPOMI PAL
v2.3.1 L2 $NO_2$ product, and diurnal variations of stratospheric $NO_2$ from the GEOS-CF v1 product. We
then calculate tropospheric $NO_2$ AMFs to convert SCDs to VCDs.

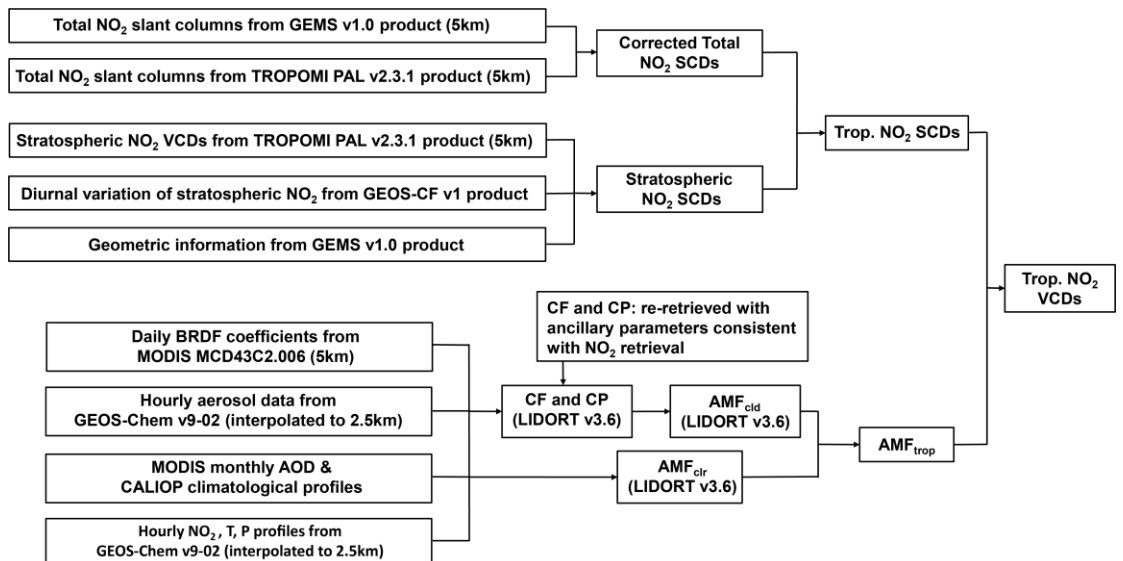


**Figure 1. Flow chart of POMINO-GEMS retrieval algorithm. The numbers in the boxes, such as 5 km, refer**
**to horizontal resolutions.**
**2.1.1 GEMS NO$_2$ and cloud data**
The GEMS instrument is on board the GK-2B satellite locating at 128.2°E over the equator (Kim
et al., 2020). The spectral wavelength range of GEMS is 300-500 nm, covering main absorption spectra
of aerosols and trace gases. The nominal spatial resolution is typically 7 km × 8 km for gases and 3.5 km
× 8 km for aerosols in the eastern and central scan domains; however, the north-south spatial resolution
can exceed 25 km in the western side. The whole field of view (FOV) covers about 20 Asian countries
within latitudes 5°S to 45°N and longitudes 80°E to 152°E. Given the variation of solar zenith angle
(SZA), there are four scan scenarios moving from east to west, including Half East (HE), Half Korea
(HK), Full Central (FC) and Full West (FW). It takes 30 minutes (for example, 00:45 – 01:15 UTC) for
GEMS to scan its full coverage during each scenario, and the next 30 minutes to transmit data to the
ground data center. The number of hourly GEMS observations per day varies from 6 in winter to 10 in
summer, corresponding to the annual movement of subsolar points relative to the Earth.
We take hourly total (stratospheric + tropospheric) NO$_2$ SCDs from the official GEMS v1.0 L2 NO$_2$
product, and convert them to 0.05° × 0.05 ° gridded data by means of an area-weighted oversampling
technique. The value of each grid cell is the mean value of pixel-based GEMS observations weighted by
the ratio of the overlap area of each pixel to the area of grid cell. We also use continuum reflectance data
(i.e., spectrally smooth reflectance from molecular and aerosol extinction as well as surface reflectance
effects) and O$_2$-O$_2$ SCDs from the official GEMS v1.0 L2 cloud product to re-calculate cloud parameters
as a prerequisite for tropospheric $NO_2$ AMF calculations. Details of the GEMS retrievals can be found
in the algorithm theoretical basis document (ATBD) (Lee et al., 2020).

**2.1.2 TROPOMI, OMI and GOME-2 $NO_2$ data**

Table S1 compares the basic information of GEMS with those of TROPOMI, OMI and GOME-2
instruments. In this study, TROPOMI data are used for derivation of POMINO-GEMS $NO_2$ VCDs, and
data from all of the three LEO instruments are used for comparison with POMINO-GEMS.
We use total $NO_2$ SCDs and stratospheric $NO_2$ VCDs from the official TROPOMI PAL v2.3.1 L2
$NO_2$ product, and convert them to $0.05° \times 0.05°$gridded data, again using an area-weighted oversampling
technique. Details of TROPOMI total SCD retrievals and stratospheric VCD calculations are given in
the TROPOMI ATBD (Van Geffen et al., 2022a). The TROPOMI PAL product is reprocessed with
TROPOMI $NO_2$ data processor v2.3.1 for the period from 1 May 2018 to 14 November 2021; it will be
replaced by the full mission reprocessing with $NO_2$ processor v2.4.0 in the future (Eskes et al., 2021).
The most important improvement in this PAL product upon the previous OFFL v1.3 is the replacement
of the FRESCO-S algorithm with the FRESCO-wide cloud retrieval algorithm, which leads to higher,
more reasonable cloud pressure (CP) estimates and substantial increases in tropospheric $NO_2$ VCDs (by
20% – 50%) over polluted regions like Eastern China in winter (Eskes et al., 2021; Van Geffen et al.,
2022b).
We use the POMINO-TROPOMI v1.2.2, OMNO2 v4 (Krotkov et al., 2019) and GOME-2 GDP 4.8
(Valks et al., 2019) tropospheric $NO_2$ VCD products to compare with POMINO-GEMS results. The
previous POMINO-TROPOMI v1 data show higher accuracy in polluted situations and improved
consistency with MAX-DOAS measurements when compared with the official TM5-MP-DOMINO
(OFFLINE) product (Liu et al., 2020). POMINO-TROPOMI v1.2.2 improves upon v1 by (1) using
tropospheric $NO_2$ SCD and CP data from the updated TROPOMI PAL v2.3.1 $NO_2$ product, (2)
interpolating the daily $NO_2$, pressure, temperature and aerosol vertical profiles from nested GEOS-Chem
(v9-02) simulations into a horizontal grid of 2.5 km x 2.5 km for subsequent tropospheric AMF
calculations, and (3) including several minor bug fixes.
We select valid satellite pixels following common practice. For the daily POMINO-TROPOMI
v1.2.2 L2 $NO_2$ product, we exclude pixels with SZA or viewing zenith angle (VZA) greater than 80°,
high albedos caused by ice or snow on the ground, quality flag values (from the TROPOMI PAL v2.3.1
product) less than 0.5 or cloud radiance fraction (CRF) greater than 50%, and then map the valid data to
a $0.05° \times 0.05°$ grid. For the daily OMNO2 v4 L2 $NO_2$ product, we exclude pixels with SZA or VZA
greater than 80°, with scene Lambert-equivalent reflectivity (LER) greater than 0.3, affected by row
anomaly (XTrackQualityFlags is not zero), marked without quality assurance (vcdQualityFlag is not an
even integer) or with CRF greater than 50%, and then map the valid data to a $0.25° \times 0.25°$ grid. For the
daily GOME-2 GDP 4.8 L2 $NO_2$ product, we exclude pixels with latitude greater than 70°, SZA greater
than 80°, failed retrieval (NO2Tropo_Flag is set to 1 or 2) or with CRF greater than 50%, and then map
the valid data to a $0.5° \times 0.5°$ grid.
**2.1.3 GEOS-CF stratospheric $NO_2$ data**
The NASA GEOS-CF system combines the Global Earth Observing System (GEOS) weather
analysis and forecasting system with GEOS-Chem v12.0.1 chemistry module (http://geoschem.org) to
provide near real-time estimates of atmospheric compositions with daily 5-day forecasts. Detailed
information of the model, including chemistry, emissions and deposition, and evaluation of the GEOS-
CF tropospheric simulation and forecast skill are presented in Keller et al. (2021). In particular, the
GEOS-Chem v12.0.1 chemistry scheme includes online stratospheric chemistry that is fully coupled with
tropospheric chemistry through the Unified tropospheric-stratospheric Chemistry eXtension (UCX)
mechanism (Eastham et al., 2014). The GEOS-CF stratospheric results are consistent with satellite
observations, albeit with notable underestimation of $NO_x$ and $HNO_3$ in the polar regions (Knowland et
al., 2022b).
The GEOS-CF outputs have a horizontal resolution of $0.25° \times 0.25°$ and a temporal resolution of 1
hour for $NO_2$ and other ancillary data used here (Knowland et al., 2022a). We convert instantaneous
stratospheric $NO_2$ volume mixing ratio in dry air at each hour (e.g., 00:00 UTC) into $0.05° \times 0.05°$
gridded vertical column densities based on estimated tropopause information in GEOS-CF v1. In Section
2.1.5, we first evaluate GEOS-CF v1 stratospheric $NO_2$ VCDs with those of TROPOMI PAL v2.3.1
product, and then calculate hourly stratospheric $NO_2$ VCDs by combining GEOS-CF v1 data for each
hour and TROPOMI PAL v2.3.1 stratospheric $NO_2$ VCD data in the early afternoon.
**2.1.4 Calculation of total $NO_2$ SCDs**
We use TROPOMI data to correct GEMS total $NO_2$ SCDs, given known issues in GEMS data.
Specifics for the $NO_2$ SCD retrieval of TROPOMI PAL v2.3.1 and GEMS v1.0 operational products are
provided in Table S2.
Figure 2a and b show the spatial distribution of monthly mean total NO$_2$ geometric column densities
(GCDs, calculated as SCDs divided by geometric AMFs) in June 2021 from TROPOMI PAL v2.3.1 and
GEMS v1.0, respectively. The horizontal resolution is 0.05° × 0.05 °. The GCDs are used to compare the
two products after removing the effect of measurement geometry. Matching for each day between hourly
GEMS observations and the TROPOMI data at the closest observation time is done to ensure temporal
compatibility. The figures show that the spatial pattern of GEMS GCDs agrees well with that of
TROPOMI, with high values over the North China Plain (NCP) and Northwestern India, as well as major
metropolitan clusters such as Seoul and the Yangtze River Delta (YRD). However, there are two
systematic problems in GEMS GCDs. First, the GEMS GCD values are abnormally high over the
northern and northwestern parts of GEMS FOV, especially over Mongolia, Qinghai, Inner Mongolia,
Xinjiang and Tibet of China. Second, west-east stripes exist over the whole domain, similar to the
spurious across-track variability issue for OMI. This stripe issue exists at all hours (Figure S1). It is likely
associated with the specific scan modes of GEMS, as well as periodically occurring bad pixels as one of
remaining calibration issues (Boersma et al., 2011; Lee et al., 2023).

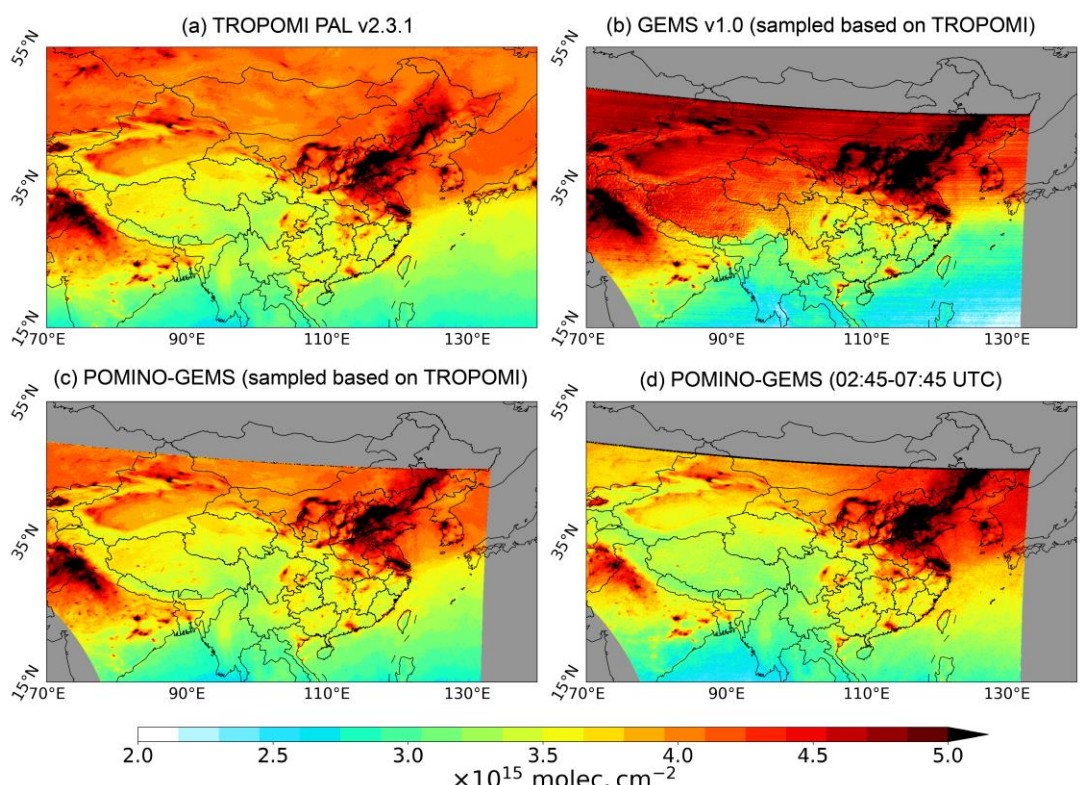


**Figure 2. Spatial distribution of monthly mean total NO$_2$ GCDs on a 0.05° × 0.05 °grid in June 2021. (a) The**
**TROPOMI PAL v2.3.1 product, (b) the official GEMS v1.0 product that spatiotemporally matches with**
**TROPOMI, (c) the corrected POMINO-GEMS product that spatiotemporally matches with TROPOMI,**
To correct the two issues in the GEMS official total $NO_2$ SCD product, we combine GEMS and
TROPOMI observations to obtain hourly $0.05° \times 0.05°$ corrected total $NO_2$ SCDs for each day using Eqs.
(1) and (2):

$$\Delta GCD = \frac{1}{n} \sum_{i=1}^{n} (GCD_{total,h_i}^{TROPOMI} - GCD_{total,h_i}^{GEMS}) \tag{1}$$

$$SCD_{total,h}^{corrected} = SCD_{total,h}^{GEMS} + \Delta GCD \times AMFgeo_h^{GEMS} \tag{2}$$

In Eqs. (1) and (2), index $h$ represents the hour of GEMS observations on each day; $h_i$ the hour
when both GEMS and TROPOMI have valid observations for the same grid cell; and $n$ the number of $h_i$.
The value of $n$ is 1 or 2 depending on the overpass times of TROPOMI. There are two steps in the
correction process. First, we calculate a geometry-independent correction map for each day using total
$NO_2$ GCDs from GEMS and TROPOMI that match spatially and temporally (Eq. (1)). We use the
absolute difference instead of a scaling factor as a simple correction. We then apply the correction to the
original GEMS total $NO_2$ SCDs at each hour on the same day, with the diurnal variation in AMF
associated with measurement geometry accounted for (Eq. (2)).
In Eq. (2), we implement a simple geometric correction (concerning SZAs and VZAs) for AMFs
instead of using the actual AMFs; the latter could account for the differences in relative azimuth angles
and other factors. Specific derivation of this assumption is given in Section 1 of the Supplement
Information (SI). The correction is assumed to be acceptable with an extra uncertainty introduced to the
total $NO_2$ SCDs, as will be further discussed in Section 3.5.
Figure 2c shows the monthly mean corrected POMINO-GEMS total $NO_2$ GCDs in June 2021 after
spatial and temporal matching with TROPOMI. The corrected GCD values in the northern GEMS FOV
are much reduced compared with those in the original GEMS data. Moreover, most stripe-like patterns
are removed in the corrected GCDs. Figure 2d is similar to Fig. 2c but for GCDs averaged over 02:45 –
07:45 UTC in June 2021. Figure S3 further compares the original GEMS and POMINO-GEMS total
$NO_2$ GCDs at each hour in JJA 2021, showing similar improvements as well. The differences between
Figure 2c and d indicate the influence of different sampling hours combined with the daily correction
map. Specifically, the correction value of each grid cell is calculated at the specific hour when both
GEMS and TROPOMI have valid observations, but this value is applied to original GEMS SCDs at all
hours.
Our correction method is done for each grid cell. We tested other correction methods by applying
the same correction value to grid cells within a $20° \times 20°$ domain, at the same latitude, or at the same
longitude. These alternative methods can reduce the high bias over the northern and northwestern GEMS
FOV to various extents, but cannot remove the stripes (not shown). We also note that our simple
correction is a temporary solution before the aforementioned systematic problems in the official GEMS
SCD retrieval are solved by improving spectral fitting. In Sections 3.3 and 3.4, we compare the diurnal
variations of tropospheric NO$_2$ VCDs based on corrected and uncorrected GEMS SCDs.
**2.1.5 Calculation of stratospheric and tropospheric NO$_2$ SCDs**
We construct a dataset of hourly stratospheric NO$_2$ SCDs at $0.05° \times 0.05°$ by using TROPOMI PAL
v2.3.1 stratospheric NO$_2$ VCDs, diurnal variation of stratospheric NO$_2$ VCDs provided by GEOS-CF v1
product, and GEMS geometric AMFs.
Figure S4 shows the comparison results between GEOS-CF v1 and TROPOMI PAL v2.3.1
stratospheric NO$_2$ VCDs in June 2021. Consistent spatial and temporal sampling is done. N is the total
number of matched $0.05° \times 0.05°$ grid cells. The stratospheric VCDs from both products vary in the range
of $2 - 5 \times 10^{15}$ molec. cm$^{-2}$, with spatiotemporal correlation of 0.99, linear regression slope of 0.99 and
normalized mean bias (NMB) of 0.02%. This consistency provides confidence on the overall reliability
of GEOS-CF stratospheric NO$_2$ data.
First, we calculate stratospheric NO$_2$ VCDs at a reference hour for each day using Eqs. (3) and (4):
$$\text{ratio}_{h_0}^h = \frac{\text{VCD}_{\text{strat},h}^{\text{GEOS-CF}}}{\text{VCD}_{\text{strat},h_0}^{\text{GEOS-CF}}} \quad (3)$$
$$\text{VCD}_{\text{strat},h_0} = \frac{1}{n} \sum_{i=1}^{n} \frac{\text{VCD}_{\text{strat},h_i}^{\text{TROPOMI}}}{\text{ratio}_{h_0}^{h_i}} \quad (4)$$
Here, Eq. (3) defines the ratio of GEOS-CF stratospheric NO$_2$ at hour $h$ to that at the reference hour
$h_0$, which is chosen to be 01:00 UTC (Figure S5). In Eq. (4), $h_i$ represents the observation time of every
TROPOMI orbit that overlaps with GEMS FOV, and $n$ the number of $h_i$ for each grid cell.
Second, we use the ratio from a given time $h$ to $h_0$ and stratospheric NO$_2$ VCDs at $h_0$ to derive
stratospheric NO$_2$ VCDs at $h$ for each day (Eq. (5)).
$$\text{VCD}_{\text{strat},h} = \text{VCD}_{\text{strat},h_0} \times \text{ratio}_{h_0}^h \quad (5)$$
Figure 3 shows the derived monthly mean stratospheric NO$_2$ VCDs at each hour in June 2021 on a
$0.05° \times 0.05°$ grid. The abrupt decease of stratospheric $NO_2$ VCDs after sunrise is caused by resumed
photochemical conversion of $NO_2$ to NO (Li et al., 2021b). There is a strong meridional gradient of
stratospheric $NO_2$ in the daytime, with the higher values in the north associated with longer lifetimes.
The stratospheric $NO_2$ increase quasi-linearly during the daytime; linear regression to the mean
stratospheric $NO_2$ VCDs over the whole domain from 01:45 to 07:45 UTC results in an increasing rate
of $(1.12\pm0.03) \times 10^{14}$ molec. $cm^{-2}$ $h^{-1}$. This result is consistent with previous work showing quasi-linear
growth in the daytime at rates of $0.5 - 2 \times 10^{14}$ molec. $cm^{-2}$ $h^{-1}$ depending on latitude and season (Li et
al., 2021b; Dirksen et al., 2011).

Finally, we use GEMS geometric AMFs to convert the stratospheric $NO_2$ VCDs to SCDs at each

hour, and then subtract them from the total SCDs to obtain tropospheric SCDs (Eqs. (6) and (7)). In the
stratosphere, the geometric AMFs are essentially the same as the actual AMFs
$$SCD_{strat,h} = VCD_{strat,h} \times AMFgeo_h^{GEMS} \qquad (6)$$
$$SCD_{trop,h}^{GEMS*} = SCD_{total,h}^{corrected} - SCD_{strat,h} \qquad (7)$$

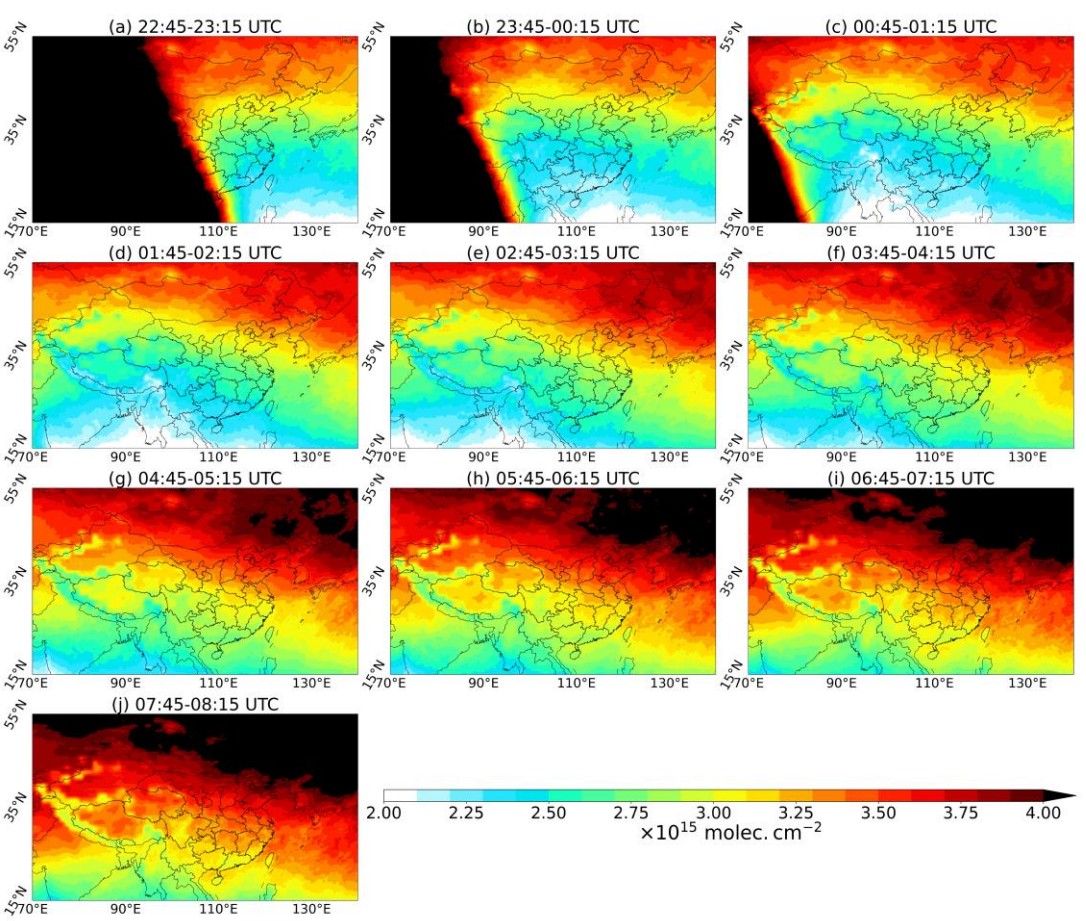


**Figure 3. Spatial distribution of POMINO-GEMS derived monthly mean stratospheric $NO_2$ VCDs at each**
**hour on a $0.05° \times 0.05°$ grid in June 2021. Note the range of the color bar is $2.0 - 4.0 \times 10^{15}$ molec. $cm^{-2}$.**

**2.1.6 Calculation of tropospheric AMFs**

Tropospheric $NO_2$ AMF is dependent on three factors as defined in Palmer et al. (2001): the viewing geometry, the scattering weights describing the sensitivity of the backscattered spectrum to the abundance of the absorber, and a priori $NO_2$ vertical profile (Eq. (8)).

$$\text{AMF} = \text{AMF}_G \int_0^{z_T} w(z)S(z)\mathrm{d}z \tag{8}$$

In Eq. (8), $\text{AMF}_G$ is the geometric AMF and a function of SZA and VZA, $w(z)$ the scattering weight at altitude $z$, $S(z)$ the normalized vertical profile of $NO_2$ number density, and $z_T$ the tropopause. Following Yang et al. (2023), we refer to $\int_0^{z_T} w(z)S(z)\mathrm{d}z$ as the scattering correction factor for discussion in Section 3.2. For tropospheric AMF calculations (Figure 1), we use a parallelized AMFv6 package driven by LIDORT version 3.6; this is similar to the one used in our previous POMINO products (Lin et al., 2014; Lin et al., 2015; Liu et al., 2019) but with modifications to adapt to the geostationary observing characteristics and high spatiotemporal resolution of GEMS. We take daily BRDF coefficients with a horizontal resolution of 5 km from the MODIS MCD43C2.006 dataset (Lucht et al., 2000) to account for the anisotropy of surface reflectance over land and coastal ocean regions , and OMLER v3 albedo over open ocean (Zhou et al., 2010; Lin et al., 2014; Liu et al., 2020). Hourly-varying aerosol parameters, a priori $NO_2$ profiles, temperature profiles and pressure profiles are interpolated from nested GEOS-Chem (v9-02) results to a horizontal resolution of 2.5 km, using the Piecewise Cubic Hermite Interpolating Polynomial (PCHIP) method. Furthermore, we deploy AOD observations from the MODIS/Aqua Collection 6.1 MYD04_L2 dataset to constrain model-simulated AOD on a monthly basis (Lin et al., 2014; Lin et al., 2015; Liu et al., 2019; Liu et al., 2020); and we use a self-constructed monthly climatological dataset of aerosol extinction profiles based on CALIOP L2 data over 2007-2015 to constrain modeled aerosol vertical profiles on a monthly climatology basis (Liu et al., 2019). We re-retrieve cloud parameters based on $O_2$-$O_2$ SCDs and continuum reflectances from the official GEMS v1.0 cloud product, using ancillary parameters consistent with those used in $NO_2$ AMF calculations. Instead of relying on a look-up table (LUT), we conduct pixel-by-pixel radiative transfer calculations with the parallelized AMFv6 package. The independent pixel approximation (IPA) is assumed for cloud-contaminated pixels as in other algorithms. Finally, we use the AMF data to convert tropospheric $NO_2$ SCDs to VCDs.

Invalid pixels in the POMINO-GEMS product are filtered based on the following criteria. We

exclude pixels with SZA or VZA greater than 80°, or with the ground covered by ice or snow. To

minimize cloud contamination, we exclude pixels with CRF greater than 50%.

**2.2 Estimation of surface NO₂ concentrations**

In order to validate satellite NO₂ products with surface concentration measurements from MEE, we

convert tropospheric NO₂ VCDs from satellite products on a $0.05° \times 0.05°$ grid to surface NO₂ mass

concentrations using GEOS-Chem simulated NO₂ vertical profiles and the box heights of the lowest

model layer (Eq. (9)).

$$C_{\text{surf}} = \text{VCD}_{\text{trop}}^{\text{SAT}} \times R^{\text{GC}} \times \frac{M}{N \times H^{\text{GC}}} \times 2 \tag{9}$$

In Eq. (9), $C_{\text{surf}}$ represents the estimated surface NO₂ mass concentration in $\mu g\ m^{-3}$, $\text{VCD}_{\text{trop}}^{\text{SAT}}$ the

satellite tropospheric VCD in molecules. $m^{-2}$, $R^{\text{GC}}$ the GEOS-Chem simulated hourly ratio of NO₂ sub-

column in the lowest layer to the total tropospheric column, $M$ the NO₂ molar mass in $\mu g\ mol^{-1}$, $N$ the

Avogadro constant, and $H^{\text{GC}}$ the box height of the lowest layer in m. The thickness of the lowest layer

of GEOS-Chem (about 130 m) is too large for the layer average NO₂ mass concentration to represent

that near the ground (Liu et al., 2018a); thus the derived concentration is multiplied by a factor of 2 to

roughly account for the vertical gradient from the height of ground instrument to the center of the model

layer. However, the constant correction factor of 2 neglects the diurnal variation of NO₂ vertical gradient,

which is related to the diurnal variation of planetary boundary layer (PBL) heights. This issue is discussed

in detail in Section 3.4.

**2.3 Ground-based MAX-DOAS measurements**

We use ground-based MAX-DOAS NO₂ measurements, together with POMINO-TROPOMI v1.2.2,

OMNO2 v4 and GOME-2 GDP 4.8 NO₂ products, to validate the POMINO-GEMS retrieval results. The

types, geolocations and observation times of MAX-DOAS stations are summarized in Table S3, and the

location of each site is shown in Figure S6. Details of each site are described in Section 2 of the SI.

Kanaya et al. (2014) and Hendrick et al. (2014) have discussed the error in MAX-DOAS NO₂ retrieval:

uncertainties from a priori aerosol and NO₂ profiles are the largest source by 10% – 14%, and the total

retrieval uncertainty is typically 12% – 17%.

To ensure sampling consistency in time, we average all valid MAX-DOAS measurements within

each observation period of GEMS (i.e., 30 minutes) for hourly comparison, and within $\pm 1.5$ h of

TROPOMI, OMI and GOME-2 overpass time for daily comparison. Following the procedures in previous studies (Lin et al., 2014; Liu et al., 2020), we exclude all matched MAX-DOAS data for which the standard deviation exceeds 20% of the mean value to minimize the influence of local events. To ensure sampling consistency in space, we select valid satellite pixels within 5 km of MAX-DOAS sites for POMINO-GEMS and POMINO-TROPOMI v1.2.2, 25 km for OMNO2 v4 and 50 km for GOME-2 GDP 4.8, and conduct spatial averaging. The Grubbs statistical test, which is used to detect outliers in a univariate data set assumed to exhibit normal distribution (Grubbs, 1950), is performed to exclude outliers in both MAX-DOAS and satellite data before comparison. Only one data pair from Fudan University site is identified as an outlier and removed (Figure S7), and we get 1348 matched hourly data pairs in total.

**2.4 Mobile-car MAX-DOAS measurements**

We use tropospheric $NO_2$ VCDs from mobile-car MAX-DOAS measurements performed by the Chinese Academy of Meteorological Sciences (CAMS) in the Three Rivers' Source region in July 2021 (Cheng et al., 2023). The Three Rivers' Source region is on the northeastern Tibetan Plateau in western China, which is isolated from massive anthropogenic activities, and hence a good place for observations of atmospheric compositions in the background atmosphere. The field campaign lasted from 18[th] to 30[th] July 2021 and included four closed-loop journeys, beginning from the meteorological bureau of the city of Xining (the Capital of Qinghai Province) to the meteorological bureau of Dari County of the Guoluo Tibetan Autonomous Prefecture, to the meteorological bureau of Yushu Tibetan Autonomous Prefecture, and then back to Xining City (Figure S6). The spectral analysis of the measurement spectra in the fitting window of 400-434 nm was implemented with the DOAS method. Sequential Fraunhofer reference spectrum (FRS) is used to derive $NO_2$ differential slant column densities (DSCDs), which are then converted to VCDs by adopting the geometric approximation method. The errors are estimated to be less than 20% at high altitudes. More detailed descriptions of instrumentation, field campaign and data retrieval are in Cheng et al. (2023).

We average all valid mobile-car MAX-DOAS measurements within each observation period of GEMS in each $0.05° \times 0.05°$ grid cell, to ensure spatiotemporal consistency. Over relatively clean areas with little human influence and biomass burning such as the Three Rivers' Source region, a large portion of $NO_2$ is located in the middle and upper troposphere, which is not accounted for in the mobile-car data

via such a DSCD-based retrieval method. Indeed, Cheng et al. (2023) showed that the official TROPOMI
$NO_2$ VCDs are higher than mobile-car data by about 40%. Considering that the diurnal variation of
middle and upper tropospheric $NO_2$ is much smaller than that in the lower troposphere, we focus on the
correlation of $NO_2$ diurnal variation between POMINO-GEMS and mobile car MAX-DOAS data.
**2.5 Ground-based MEE $NO_2$ measurements**
We use hourly surface $NO_2$ mass concentration measurements from the MEE air quality monitoring
network. By 2021, more than 2000 MEE stations across China have been established, providing hourly
observations for $NO_2$ and five other air pollutants. Most stations are in urban or suburban areas.
The spatial distribution of all MEE sites in the GEMS FOV is shown in Figure S8a, and that of MEE
sites over urban, suburban and rural regions are shown in Figure S8b–d, respectively. The classification
of sites is based on Tencent user location data with a horizontal resolution of 0.05° × 0.05 °for every 0.5
second from 31 August to 30 September 2021 (Figure S8e), adopted from previous work (Kong et al.,
2022a). Here, urban MEE sites are defined as where the mean location request times is larger than 50
times per second, suburban sites refer to 5-50 times per second, and rural sites refer to less than 5 times
per second. The number of sites for urban, suburban and rural sites are 808, 554 and 71, respectively.
At MEE sites, molybdenum catalyzed conversion from $NO_2$ to NO and subsequent
chemiluminescence measurement of NO is done to estimate $NO_2$ concentrations. The heated
molybdenum catalyst has low chemical selectivity, leading to strong interference from other oxidized
nitrogen species such as nitric acid ($HNO_3$) and peroxyacetyl nitrate (PAN). Therefore, MEE data tend
to overestimate the actual $NO_2$ concentrations, with the extent of overestimation about 10% – 50%
(Boersma et al., 2009; Liu et al., 2018a). The overestimation is dependent on the oxidation level of $NO_x$,
but is currently unclear for each site and hour.
To compare with satellite-derived surface $NO_2$ concentration data, we average over all valid MEE
sites in each 0.05° × 0.05 °grid cell to generate gridded MEE $NO_2$ data for each hour. To ensure sampling
consistency for each day, we average MEE observations at two consecutive hours to match GEMS hourly
observations – for example, we match the mean value of MEE $NO_2$ concentrations in 13:00 – 14:00 and
14:00 – 15:00 local solar time (LST) with the GEMS $NO_2$ in 13:45 – 14:15 LST. We also match MEE
observations over the period 13:00 – 14:00 LST with TROPOMI-derived and OMI-derived surface $NO_2$,
and 9:00 – 10:00 LST with GOME-2-derived surface $NO_2$.

## 3. Results and discussion

### 3.1 POMINO-GEMS tropospheric NO₂ VCDs

Figure 4 shows mean POMINO-GEMS tropospheric $NO_2$ VCDs at each hour on a 0.05° × 0.05 ° grid in JJA 2021. High values of tropospheric $NO_2$ columns (> 10 × $10^{15}$ molec. cm$^{-2}$) are evident over populous regions such as South Korea, central and eastern China, and northern India. Clear hotspot signals reveal intense $NO_x$ emissions over city clusters such as Beijing-Tianjin-Hebei (BTH), Yangtze River Delta (YRD), Pearl River Delta (PRD) and Seoul Metropolitan Area (SMA), as well as isolated megacities such as Osaka and Nagoya in Japan, Chengdu and Urumqi in China, and New Delhi in India. Tropospheric $NO_2$ VCDs are much lower (< 1 × $10^{15}$ molec. cm$^{-2}$) over most of western China and the open ocean, due to low anthropogenic and natural emissions.

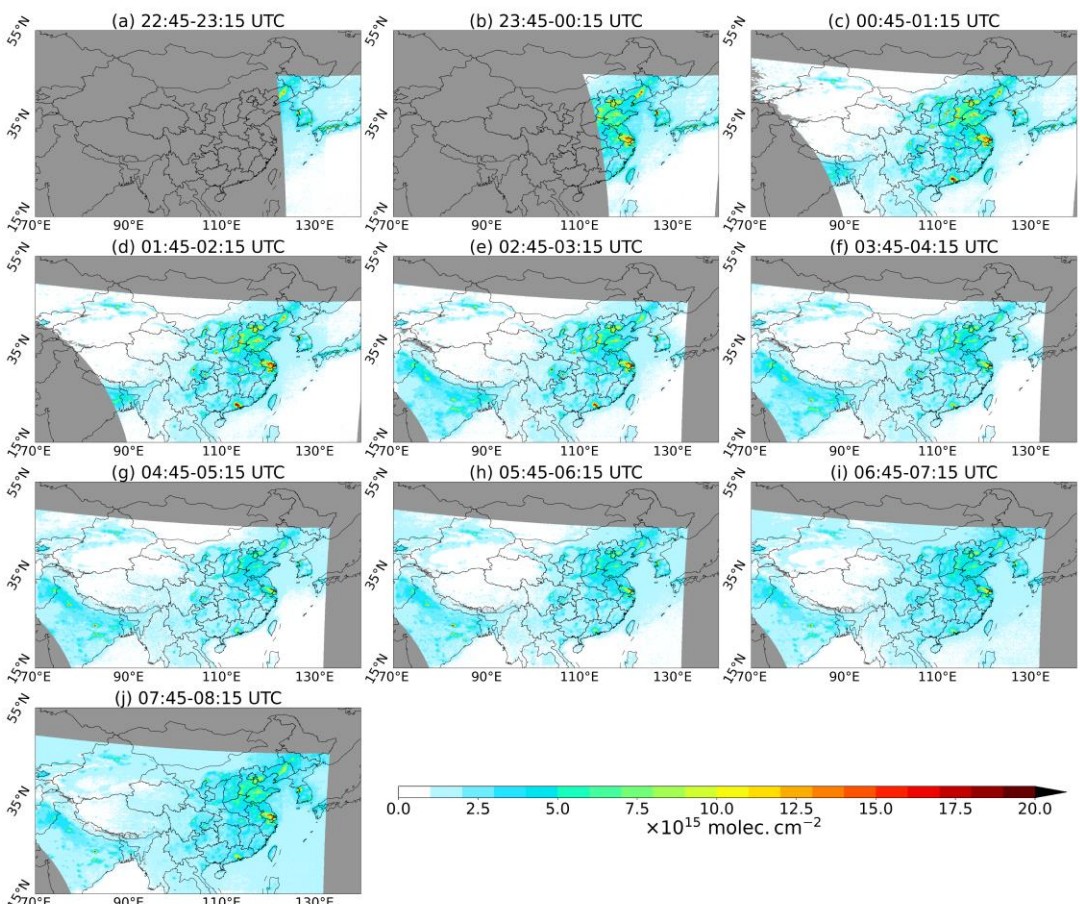

**Figure 4. Spatial distribution of POMINO-GEMS tropospheric NO₂ VCDs at each hour on a 0.05° × 0.05 ° grid in JJA 2021. The regions in grey mean there are no valid observations.**

Figure 5a-c present $NO_2$ VCDs in the morning, noon and afternoon in JJA 2021 for eastern China. Data are averaged in 22:45 – 01:45 UTC (06:45 – 09:45 Beijing Time, BJT), 02:45 – 04:45 UTC (10:45

– 12:45 BJT) and 05:45 – 07:45 UTC (13:45 – 15:45 BJT) to represent the morning, noon and afternoon,
respectively. In the morning (Figure 5a), there are clear city signals with high $NO_2$ values, reflecting
abundant $NO_x$ emissions from traffic. The spatial gradients of $NO_2$ from urban centers to outskirts are
very strong. However, these spatial gradients are greatly reduced in the noon and afternoon (Figure 5b
and c). For example, the differences of tropospheric $NO_2$ VCDs between the urban center of Xi'an
(108.93°N, 34.27°E) and its surrounding areas (within 50 km) are reduced from about $8 \times 10^{15}$ molec.
$cm^{-2}$ in the morning to about $4 \times 10^{15}$ molec. $cm^{-2}$ at noon, and then to below $2 \times 10^{15}$ molec. $cm^{-2}$ in the
afternoon. This is likely due to chemical loss of traffic-associated $NO_2$, increased emissions from other
sectors (e.g., industry), and/or enhanced horizontal transport smearing the spatial gradient.

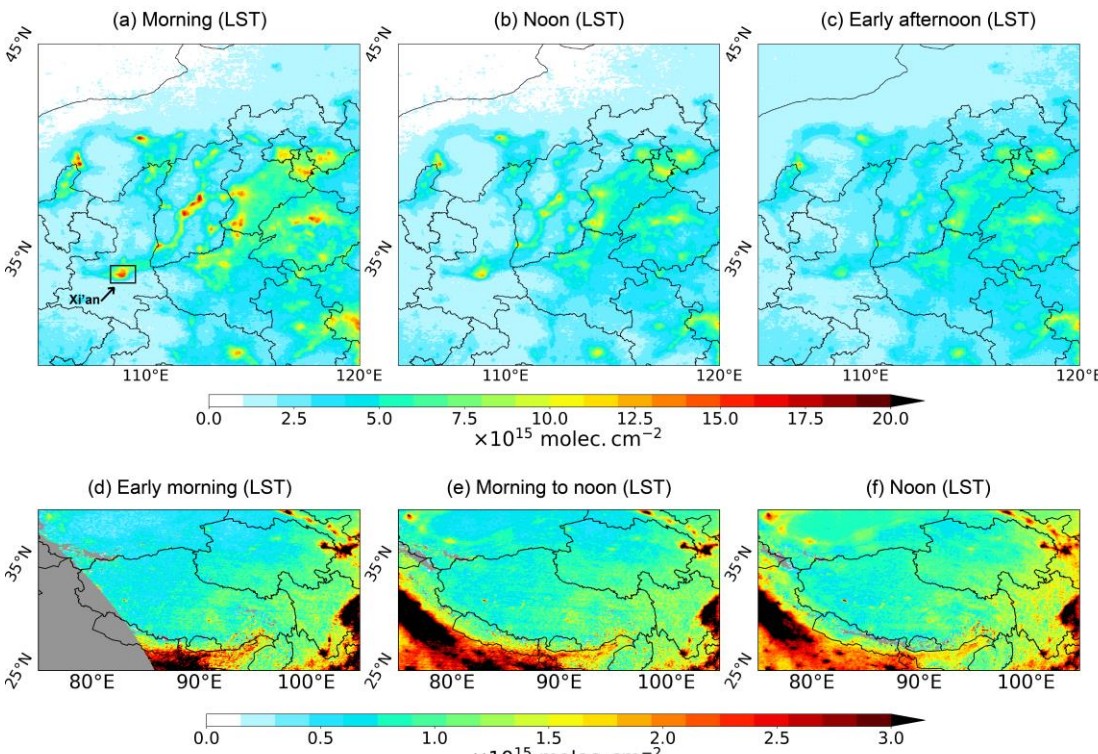

**Figure 5. Spatial distribution of three-hour-mean POMINO-GEMS tropospheric $NO_2$ VCDs in JJA 2021 on**
**a 0.05° × 0.05 °grid. The first row is for eastern China in the (a) morning (22:45 – 01:45 UTC), (b) noon**
**(02:45 – 04:45 UTC) and (c) afternoon (05:45 – 07:45 UTC). The second row is for western China in the (d)**
**early morning (00:45 – 01:45 UTC), (e) morning to noon (02:45 – 04:45 UTC) and (f) noon (05:45 – 07:45**
**UTC). The regions in grey mean there are no valid observations.**
Over western China with low tropospheric $NO_2$ VCDs (Figure 5d-f), there is a gradual increase of
tropospheric $NO_2$ by about $1 \times 10^{15}$ molec. $cm^{-2}$ from the early morning to noon. This increase is likely
dominated by biogenic $NO_x$ emissions that are sensitive to sunshine intensity and surface temperature
(Kong et al., 2022b; Weng et al., 2020; Kong et al., 2023). Future studies are needed to understand the
exact causes.
Figure 6 shows the diurnal variation of POMINO-GEMS tropospheric $NO_2$ VCDs over six different
region groups in the GEMS FOV. The six groups are defined based on the levels of mean POMINO-
GEMS tropospheric $NO_2$ VCDs at 12:00 LST in JJA 2021 ($VCD_{12:00\ LST}$), and their spatial distributions
are also shown in each panel. We convert the observation time from UTC to LST for each time zone in
this domain (+5 time zone: 70°E – 82.5° E; +6 time zone: 82.5°E – 97.5°E; +7 time zone: 97.5°E –
112.5°E; +8 time zone: 112.5°E – 127.5°E; +9 time zone: 127.5°E – 140°E), and show the $NO_2$ diurnal
variations in each time zone with different colors. For low $NO_2$ situations ($VCD_{12:00\ LST} \leq 2 \times 10^{15}$ molec.
$cm^{-2}$), $NO_2$ grow in the morning time in +5 and +6 time zones but not in other time zones. Over high
$NO_2$ situations ($VCD_{12:00\ LST} > 8 \times 10^{15}$ molec. $cm^{-2}$, in cities and suburban areas), $NO_2$ in all time zones
exhibit a minimum around noontime and a morning peak at 09:00 – 10:00 LST, consistent with previous
findings for specific polluted locations (Boersma et al., 2008; Boersma et al., 2009; Li et al., 2021a;
Ghude et al., 2020; Herman et al., 2019; Biswas and Mahajan, 2021). In all groups and time zones,
tropospheric $NO_2$ VCDs grow from noon to the afternoon.
The $NO_2$ diurnal variations are related to multiple driving factors. Different sources with distinctive
diurnal patterns dominate the $NO_x$ emissions over different regions. Lightning and biogenic activities are
the major emission sources over low $NO_2$ land areas, and they tend to intensify with temperature and
radiation in the daytime. Anthropogenic emissions are dominant over polluted cities and suburban areas,
where the traffic emissions tend to peak in the mid-morning and late afternoon (Jing et al., 2016; Liu et
al., 2018b; Naiudomthum et al., 2022). In addition, the photochemistry plays an important role. $NO_2$ is
in chemical balance with NO, and the ratio of $NO_2$ and NO depends on radiation, ozone and peroxyl
radicals. $NO_x$ is oxidized to nitric acid and organic nitrates by radicals in the daytime, the level of which
depends on radiation, ozone and volatile organic compounds. Thus the lifetime of $NO_2$ reaches the
minimum value around noon, i.e., a few hours in summer. Furthermore, atmospheric transport also affects
the diurnal variation of $NO_2$ at high-value places (e.g., cities) and their surroundings. Further studies are
needed to determine the exact causes of $NO_2$ diurnal variations at individual places.

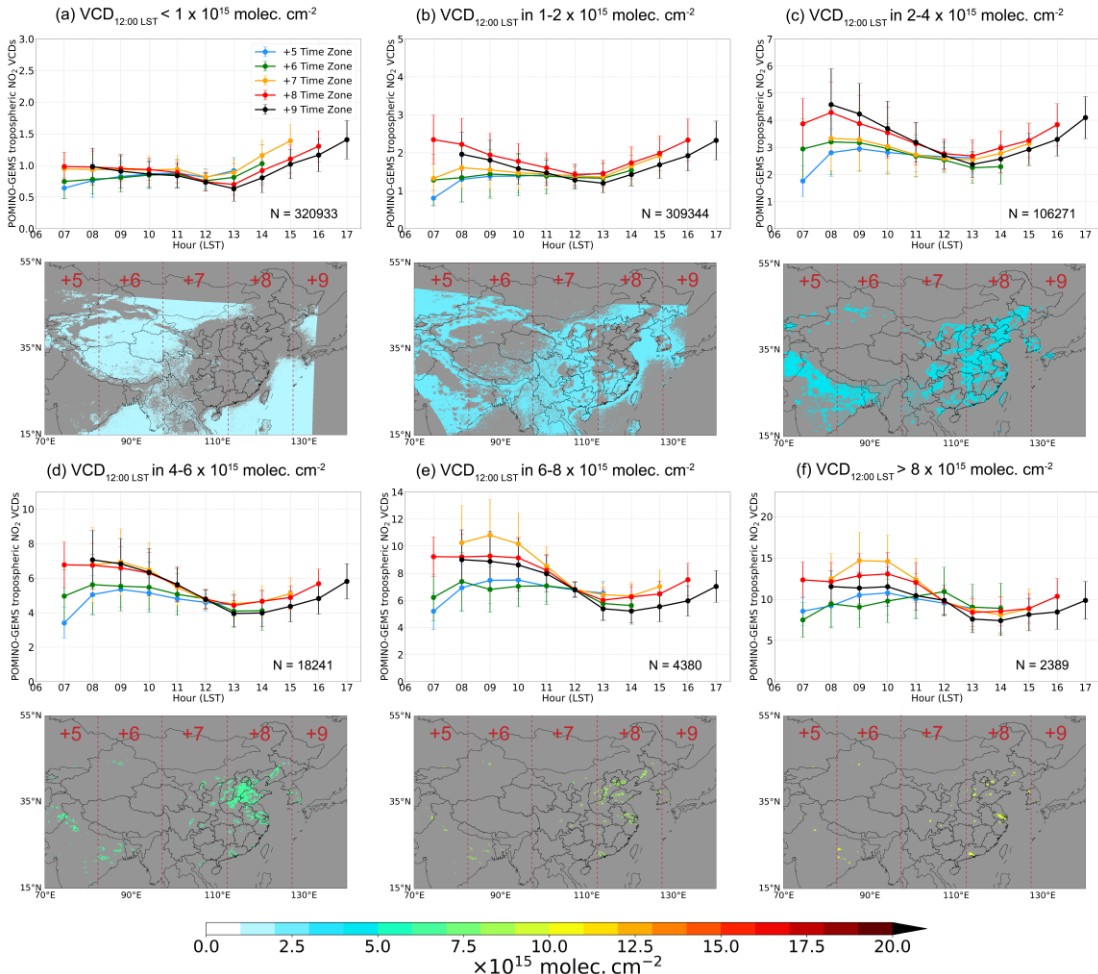


**Figure 6. POMINO-GEMS NO₂ diurnal variations for six region groups classified based on mean POMINO-GEMS tropospheric NO₂ VCDs at 12:00 LST in JJA 2021 (VCD$_{12:00\ LST}$). (a) VCD$_{12:00\ LST}$ less than $1 \times 10^{15}$ molec. cm$^{-2}$; (b) VCD$_{12:00\ LST}$ in $1 - 2 \times 10^{15}$ molec. cm$^{-2}$; (c) VCD$_{12:00\ LST}$ in $2 - 4 \times 10^{15}$ molec. cm$^{-2}$; (d) VCD$_{12:00\ LST}$ in $4 - 6 \times 10^{15}$ molec. cm$^{-2}$; (e) VCD$_{12:00\ LST}$ in $6 - 8 \times 10^{15}$ molec. cm$^{-2}$ and (f) VCD$_{12:00\ LST}$ larger than $8 \times 10^{15}$ molec. cm$^{-2}$. In each panel, different colors denote the NO₂ diurnal variation in different time zones. N denotes the total number of valid $0.05° \times 0.05°$ grid cells in each region. The error bars denote the standard deviation of tropospheric NO₂ VCDs at each hour in each time zone.**

## 3.2 Comparison with POMINO-TROPOMI v1.2.2, OMNO2 v4 and GOME-2 GDP 4.8 NO₂ VCD products

Figure 7a and b show the POMINO-GEMS and POMINO-TROPOMI v1.2.2 tropospheric NO₂ VCDs, respectively, on a $0.05° \times 0.05°$ grid averaged over JJA 2021. Cloud screening is implemented based on the CRFs from each product. To ensure temporal compatibility, matching between hourly GEMS observations and the TROPOMI data at the closest observation time is done for each day. Overall, POMINO-GEMS agrees well with POMINO-TROPOMI with a spatial correlation coefficient of 0.98, a linear regression slope of 1.18 and a small positive NMB of 4.9% (Figure 7c). Regionally, POMINO-

GEMS VCDs are higher than those of POMINO-TROPOMI v1.2.2 over eastern China, most India and
northwestern GEMS FOV, but smaller over western China and the oceans (Figure 7a, b; see Figure S9c
and d for differences plots). These differences are related to tropospheric $NO_2$ AMFs and SCDs. Detailed
discussion is given in Section 3 of the SI.

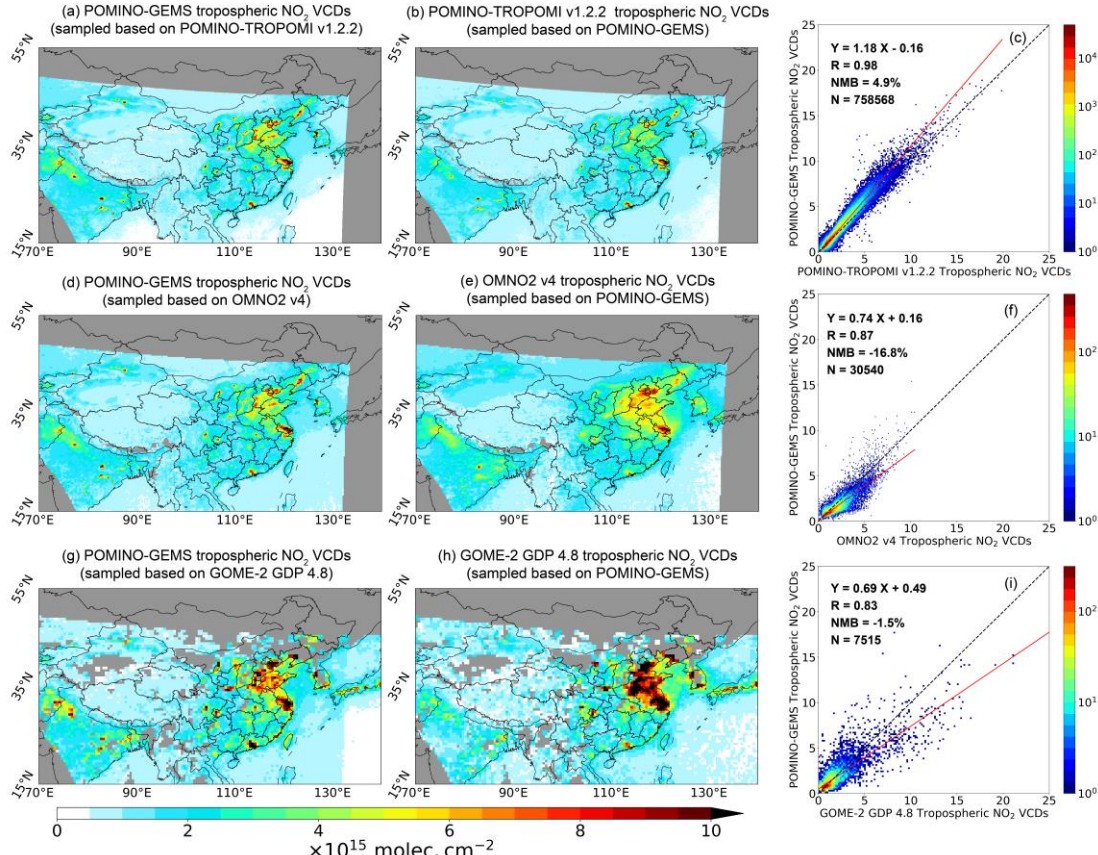


**Figure 7. Comparison between POMINO-GEMS and other products for tropospheric NO₂ VCDs in JJA**
**2021. (a-b) Between POMINO-GEMS and POMINO-TROPOMI v1.2.2 on a 0.05° × 0.05 °grid, (d-e)**
**between POMINO-GEMS and OMNO2 v4 on a 0.25° × 0.25 °grid, and (g-h) between POMINO-GEMS and**
**GOME-2 GDP 4.8 on a 0.5° × 0.5 °grid. (c), (f) and (i) are respective scatterplots, in which the colors**
**represent data density. The regions in grey mean there are no valid observations.**
Figure 7d-f and g-i show the comparison results of POMINO-GEMS tropospheric $NO_2$ VCDs with
OMNO2 v4 on a 0.25° × 0.25° grid and GOME-2 GDP 4.8 on a 0.5° × 0.5 °grid averaged over JJA 2021,
respectively. POMINO-GEMS $NO_2$ VCDs exhibit good spatial consistency with the two independent
products ($R$ = 0.87 and 0.83), although with slightly lower values than OMNO2 v4 (by 16.8%) and
GOME-2 GDP 4.8 (by 1.5%). These VCD differences are expected, considering the differences in the
retrieval algorithm. For example, the POMINO-GEMS algorithm implements explicit aerosol
corrections in the radiative transfer calculation, while OMNO2 v4 and GOME-2 GDP 4.8 treat aerosols
as "effective clouds". POMINO-GEMS accounts for the anisotropy of surface reflectance by adopting
MODIS BRDF coefficients, whereas OMNO2 v4 and GOME-2 GDP 4.8 use geometry-dependent and
regular LER, respectively. The horizontal resolution of a priori $NO_2$ profiles in POMINO-GEMS is 25
km (and interpolated to 2.5 km), $1° \times 1.25°$ in OMNO2 v4 and $1.875° \times 1.875°$ in GOME-2 GDP 4.8
(Krotkov et al., 2019; Valks, 2019).
Based on comparisons with POMINO-TROPOMI v1.2.2, OMNO2 v4 and GOME-2 GDP 4.8 $NO_2$
VCDs, we conclude that POMINO-GEMS $NO_2$ columns show good agreement with LEO satellite data,
with lower values by 20% at most.

**3.3 Validation with MAX-DOAS $NO_2$ VCD measurements**

The scatterplot in Figure 8a compares POMINO-GEMS tropospheric $NO_2$ VCDs in JJA 2021 at all
GEMS observation hours with matched ground based MAX-DOAS measurements at nine sites.
POMINO-GEMS correlates with MAX-DOAS ($R = 0.66$) with a small negative bias (NMB = −11.1%).
The linear regression shows a slope of 0.51 and intercept of $3.34 \times 10^{15}$ molec. cm$^{-2}$, reflecting
underestimation of POMINO-GEMS tropospheric $NO_2$ VCDs on high-$NO_2$ days.

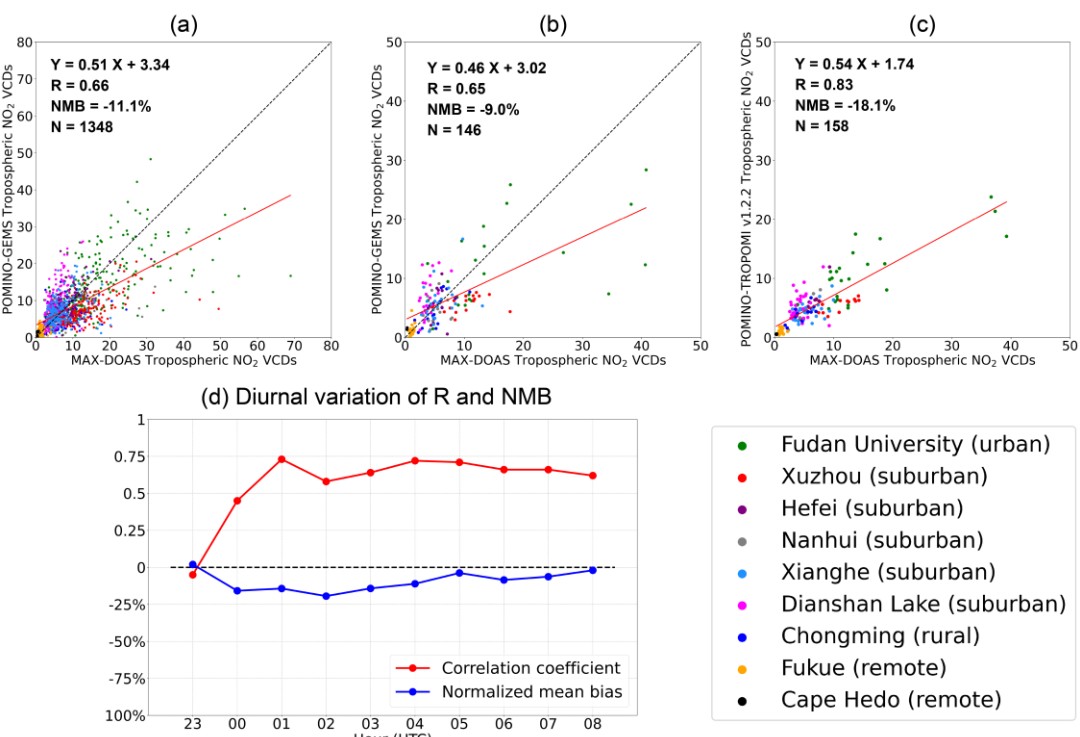

**Figure 8. Evaluation of satellite $NO_2$ VCD data using ground-based MAX-DOAS measurements. (a)**
**Scatterplot for tropospheric $NO_2$ VCDs ($\times 10^{15}$ molec. cm$^{-2}$) between MAX-DOAS and POMINO-GEMS at**
**all GEMS observation hours in JJA 2021. Each data pair denotes an hour. (b-c) Scatterplots for**
**tropospheric $NO_2$ VCDs ($\times 10^{15}$ molec. cm$^{-2}$) in JJA 2021 (b) between MAX-DOAS and POMINO-GEMS at**
**13:45 – 14:15 LST and (c) between MAX-DOAS and POMINO-TROPOMI v1.2.2. Each data pair denotes a**
**day. Each MAX-DOAS station is color-coded. (d) Diurnal variations of spatiotemporal correlation**

**coefficients and NMBs of POMINO-GEMS tropospheric NO$_2$ VCDs relative to ground-based MAX-DOAS**
**data.**

Figure 8b-c further use MAX-DOAS measurements to evaluate POMINO-GEMS and POMINO-

TROPOMI v1.2.2 tropospheric NO$_2$ VCDs at the overpass time of TROPOMI. In Figure 8b, POMINO-
GEMS data at 13:45 – 14:15 LST are used to match the overpass time of TROPOMI. The POMINO-
TROPOMI product is evaluated in the context of understanding the relative performance of POMINO-
GEMS. Each data point represents a day. Figure 8b-c show that the day-to-day variability of MAX-
DOAS measurements is well captured by POMINO-TROPOMI v1.2.2 ($R = 0.83$), but less so by
POMINO-GEMS ($R = 0.65$). Linear regression results show an underestimate of tropospheric NO$_2$ VCDs
in POMINO-TROPOMI v1.2.2 (NMB = −18.1%), as also found in previous studies (Liu et al., 2020).
POMINO-GEMS exhibits a small bias (NMB = −9.0%), but station-dependent performance is apparent.
At the two remote sites of Fukue and Cape Hedo with low NO$_2$, POMINO-GEMS NO$_2$ columns are
higher than those of MAX-DOAS measurements. At the other sites, the data pairs are more scattered and
located both above and below the 1:1 line, resulting in a small NMB.

Figure 8d shows the NMBs and correlation coefficients of POMINO-GEMS NO$_2$ VCDs relative to

ground-based MAX-DOAS data at each hour. The NMBs are negative at most hours except 23:00 UTC
(07:00 BJT). The negative NMBs reach a maximum of about 20% at 02:00 UTC (10:00 BJT), and
decrease to less than 10% in the afternoon. The correlation coefficients are modest or high (0.45 – 0.73)
at most hours, with the exception at the first hour which is likely due to few valid data (N = 17).

Figure 9 compares the diurnal variation of tropospheric NO$_2$ VCDs between POMINO-GEMS and

MAX-DOAS at eight stations. At each site, NO$_2$ values are averaged in JJA 2021 at each hour for
comparison, and the number of valid days for each hour is also shown. The Cape Hedo site is not included
because there are few valid MAX-DOAS data points at each hour. Figure 10a-f show that at the urban
and suburban sites, MAX-DOAS NO$_2$ (black lines) peaks in the mid-to-late morning, declines towards
the minimum values at noon around 13:00 LST, and then gradually increases in the afternoon. Strong
correlation of NO$_2$ diurnal variation between POMINO-GEMS (red solid lines) and MAX-DOAS is
found at Xuzhou ($R = 0.82$), Hefei ($R = 0.96$), Fudan University ($R = 0.84$), Nanhui ($R = 0.79$) and
Xianghe ($R = 0.94$). At the Dianshan Lake site, POMINO-GEMS NO$_2$ columns increase but MAX-
DOAS data decrease from 08:00 to 09:00 LST, resulting in a lower correlation coefficient ($R = 0.60$). At
Chongming and Fukue sites, MAX-DOAS NO$_2$ shows a peak in the morning without evident increase in

the early afternoon, but this diurnal pattern is not fully captured by POMINO-GEMS. At Fukue,

POMINO-GEMS NO$_2$ exhibit abrupt changes at 12:00 and 13:00 LST due to few valid data.

In addition, comparison of POMINO-GEMS diurnal variation with NO$_2$ data from GOME-2 in the

morning and OMI and TROPOMI in the early afternoon shows good agreement at Hefei, Nanhui,

Dianshan Lake, Chongming and Fukue sites. The differences between POMINO-GEMS to MAX-DOAS

NO$_2$ VCDs are comparable or smaller than those between LEO satellite and MAX-DOAS NO$_2$ VCDs.

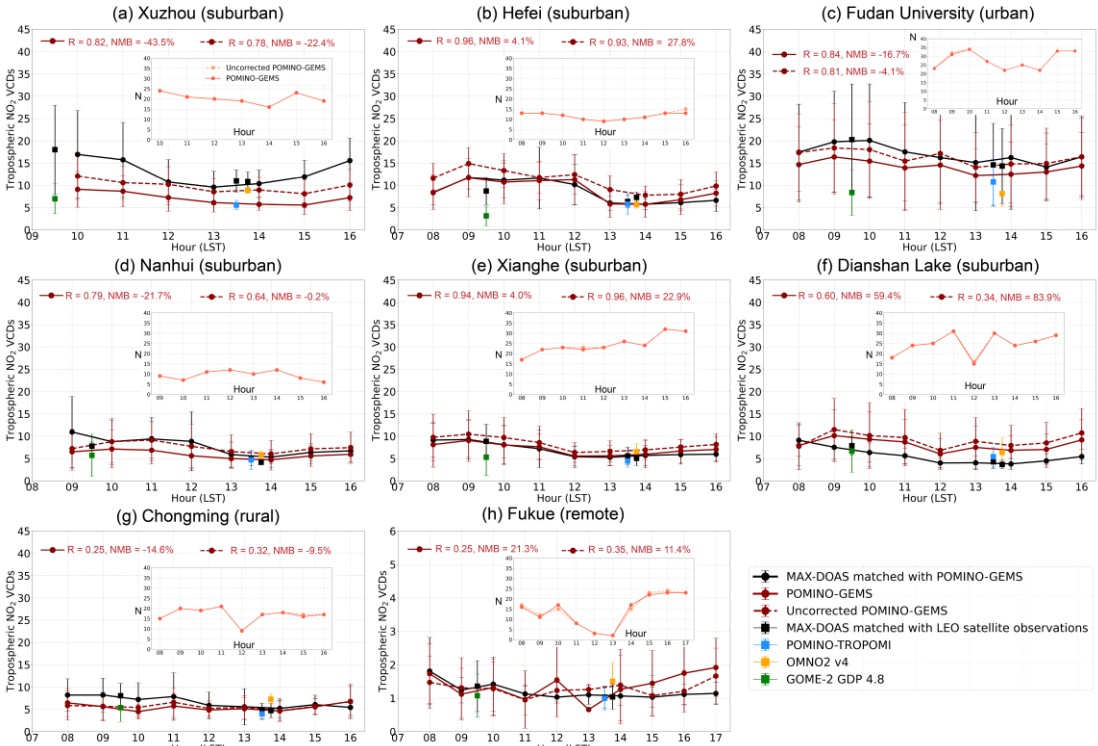

**Figure 9. Diurnal variation of hourly tropospheric NO$_2$ VCDs ($\times$ 10$^{15}$ molec. cm$^{-2}$) of MAX-DOAS (black lines), and POMINO-GEMS with TROPOMI correction (red solid lines). and re-calculated POMINO-GEMS without TROPOMI correction (red dashed lines) at eight sites in JJA 2021. The error bars denote the standard deviation of MAX-DOAS and POMINO-GEMS NO$_2$ at each hour, respectively. Diurnal correlation and all-hour-mean NMB of POMINO-GEMS against MAX-DOAS data are shown. The number of valid days for each hour is also presented. The black squares with an error bar represent the mean value and standard deviation of MAX-DOAS tropospheric NO$_2$ VCDs matched with POMINO-TROPOMI v1.2.2 (blue squares), OMNO2 v4 (orange squares) and GOME-2 GDP 4.8 (green squares), respectively.**

As we use TROPOMI total NO$_2$ SCDs to correct those of GEMS, this may influence the NO$_2$ diurnal

variation of original GEMS observations. Thus we also compare MAX-DOAS data with re-calculated

POMINO-GEMS tropospheric NO$_2$ VCDs without correction in total SCDs (red dashed lines in Figure

9). Compared to our default POMINO-GEMS data (with correction), excluding the correction leads to

lower diurnal correlation coefficients at Xuzhou, Hefei, Fudan University, Nanhui and Dianshan Lake,

but higher correlation coefficients at Xianghe, Chongming and Fukue. Excluding the correction increases
the NMB at three sites but decreases the NMB at five sites. We conclude that at these eight sites (in the
eastern areas), no significant influence on the diurnal variation of POMINO-GEMS tropospheric $NO_2$
VCDs is brought in through TROPOMI-based correction for total $NO_2$ SCDs.

Figure 10 compares the diurnal variations between POMINO-GEMS and mobile-car MAX-DOAS

tropospheric $NO_2$ VCD data in the Three Rivers' Source region on the Tibetan Plateau. Results of
POMINO-GEMS with and without total SCD correction are shown in the red solid and dashed lines,
respectively. Mobile-car MAX-DOAS data show an evident decrease of tropospheric $NO_2$ VCDs from
the morning to noon with little change thereafter. Such $NO_2$ diurnal patterns reflect the spatial and
temporal variations of tropospheric $NO_2$ along the driving route. The high $NO_2$ values with large standard
deviation at 09:00 BJT is due to enhanced pollution and variability in the morning when the car is in or
near the Xining city. The $NO_2$ diurnal variations of POMINO-GEMS with correction correlate well with
those of mobile-car MAX-DOAS data ($R$ = 0.81). In contrast, POMINO-GEMS without total SCD
correction exhibits much poorer correlation with mobile-car MAX-DOAS data, due to the erroneous
increase in the afternoon.

Overall, the validation results with independent ground-based and mobile-car MAX-DOAS

measurements provide confidence on the general characteristics of POMINO-GEMS $NO_2$ diurnal
variations.

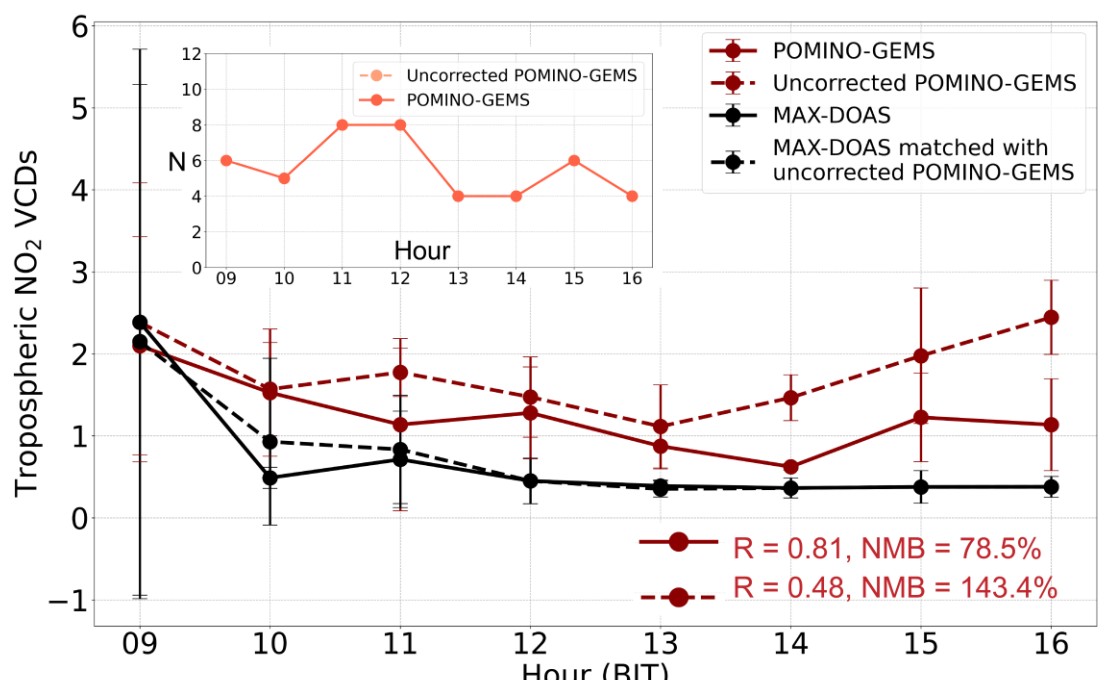


**Figure 10. Diurnal variation of hourly mean tropospheric NO₂ VCDs (× 10¹⁵ molec. cm⁻²) of mobile-car**
**MAX-DOAS and POMINO-GEMS in the Three Rivers' Source region. The black solid lines denote MAX-**
**DOAS data that spatiotemporally match with POMINO-GEMS with total SCD correction (red solid lines).**
**The black dashed lines denote MAX-DOAS data that spatiotemporally match with POMINO-GEMS**
**without correction (red dashed lines). The error bars denote the standard deviation of MAX-DOAS and**
**POMINO-GEMS NO₂ at each hour during the field campaign, respectively. Values for diurnal correlation**
**and mean NMB of POMINO-GEMS relative to MAX-DOAS are shown. The number of days with valid**
**data for each hour is also presented.**
**3.4 Validation with surface NO₂ concentration measurements from MEE**
The scatterplot in Figure 11a compares surface NO₂ concentrations derived from POMINO-GEMS
with MEE measurements at all hours. POMINO-GEMS derived surface NO₂ concentrations show good
agreement with MEE measurements in terms of spatiotemporal correlation ($R = 0.78$) and bias (NMB =
−26.3%), but are higher than those of MEE at some high-value situations, which mainly occur over the
YRD region (Figure S14). These differences reflect errors in POMINO-GEMS NO₂ VCDs, in the
conversion from tropospheric VCDs to surface concentrations, and in MEE data (due to potential
contamination by nitric acid and organic nitrates (Liu et al., 2018a)).

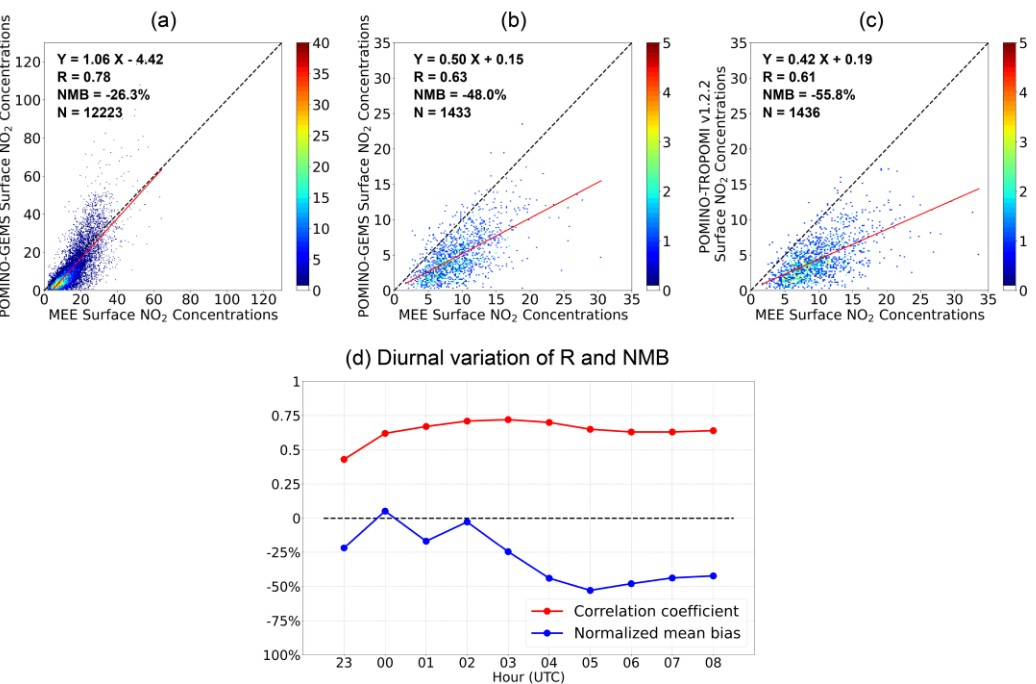


**Figure 11. Evaluation of satellite-derived surface NO₂ concentrations (µg m⁻³) using MEE measurements in**
**JJA 2021. (a) Scatterplot for MEE and POMINO-GEMS at all GEMS observation hours averaged over all**
**days in JJA 2021. (b) Scatterplot for MEE and POMINO-GEMS at 13:45 – 14:15 LST. (c) Scatterplot for**
**MEE and POMINO-TROPOMI v1.2.2. The color bar represents the data density. (d) Diurnal variations of**
**spatiotemporal correlation coefficients and NMBs of POMINO-GEMS derived surface NO₂ concentrations**
**relative to MEE measurements.**
Figure 11b-c show validation results for satellite-derived surface $NO_2$ concentrations with MEE
measurements at the overpass time of TROPOMI (i.e., early afternoon). Here, each data pair denotes a
MEE site. POMINO-GEMS results at 13:45 – 14:15 LST are used to match the overpass time of
TROPOMI data. Overall, both satellite-based datasets show good spatial correlation with MEE
measurements ($R = 0.63$ and $0.61$). POMINO-GEMS exhibits higher linear regression slope (0.50) with
smaller NMB ($-48.0\%$). The values of satellite data are lower than those from MEE, especially in the
afternoon (Figure 11d). This is in part because of the aforementioned contamination issues in MEE data,
which becomes severer in the afternoon as the air gets more aged throughout the daytime.
Figure 12a examines the diurnal variation of surface $NO_2$ concentrations averaged over JJA 2021 at
all sites. The MEE data show a smooth and monotonic decline from the early morning to the early
afternoon, with a slight increase beginning at 15:00 LST. This diurnal pattern differs from those seen in
ground-based MAX-DOAS VCD data (Figure 9), due to the difference in sampling size between MEE
and MAX-DOAS, the diurnal variation of $NO_2$ vertical distribution that affects the relationship between
surface and columnar $NO_2$, as well as the insensitivity of $NO_2$ columns to changes in PBL heights.
POMINO-GEMS derived surface $NO_2$ concentrations show similar diurnal variations to those of MEE
($R = 0.97$), although with a peak at 10:00 LST and a gradual increase beginning at 14:00 LST. The
discrepancies between POMINO-GEMS and MEE surface $NO_2$ concentrations at different hours are
likely caused by the assumed constant correction factor of 2 to account for the vertical gradient of $NO_2$
from the height of ground instrument to the center of the first model layer (Section 2.2). In the morning
when the PBL is low, most $NO_2$ molecules are near the ground and the vertical gradient of $NO_2$ over
polluted regions is the largest in the daytime, so the factor of 2 may lead to underestimation of derived
surface $NO_2$ concentrations. In contrast, in the afternoon, the PBL mixing is much stronger and the
vertical gradient of $NO_2$ is much smaller, thus the factor of 2 may lead to overestimated surface $NO_2$
concentrations. Note that the consistency between POMINO-GEMS and MEE data does not depend on
the total SCD correction (Table S4).
To quantify the influences of the diurnal variation of hourly column-to-surface ratio from GEOS-
Chem simulations, we compare the MEE measurements with POMINO-GEMS derived surface $NO_2$
concentrations using daily column-to-surface ratio (Figure S15). As expected, POMINO-GEMS derived
$NO_2$ concentrations show a similar diurnal variation as the tropospheric $NO_2$ VCDs do, with two peaks

645 in the mid-morning and afternoon, and a minimum at noon. The temporal correlation coefficient with

646 MEE is only about 0.23. Thus it is more reasonable to use hourly ratio for comparison with MEE

647 measurements, as done in our study.

648  To further test the reliability of our VCD-to-surface-concentration conversion method (Eq. (9)), we

649 apply the same method to MAX-DOAS $NO_2$ VCDs and compare the resulting surface $NO_2$

650 concentrations with MEE data. As shown in Figure S16, the diurnal variation of MAX-DOAS derived

651 surface $NO_2$ concentrations correlates well with that of MEE measurements ($R = 0.96$), in support of our

652 conversion method.

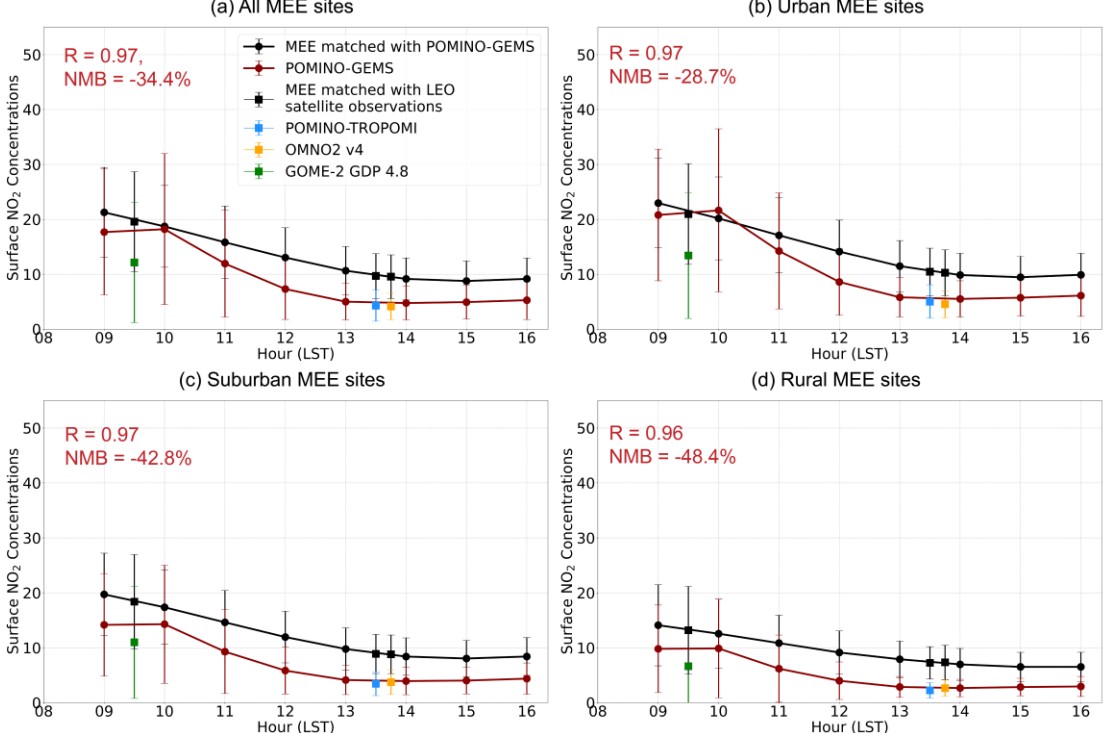

653

654 **Figure 12. Diurnal variation of hourly surface $NO_2$ concentrations (µg m$^{-3}$) of MEE (back lines) and**

655 **POMINO-GEMS (red lines) in JJA 2021. (a) At all MEE sites, (b) at urban sites, (c) at suburban sites and**

656 **(d) at rural sites. The error bars denote the standard deviation of MEE and POMINO-GEMS derived**

657 **surface $NO_2$ concentrations at each hour in JJA 2021, respectively. Diurnal correlation and mean NMB of**

658 **POMINO-GEMS relative to MEE are also listed. The black squares with an error bar represent the mean**

659 **value and standard deviation of MEE data matched with POMINO-TROPOMI v1.2.2 (blue squares),**

660 **OMNO2 v4 (orange squares) and GOME-2 GDP 4.8 (green squares), respectively.**

661  Figure 12b-d show the comparison of $NO_2$ diurnal variations for different groups of MEE sites. The

662 diurnal variations of POMINO-GEMS derived surface $NO_2$ concentrations show similar characteristics

663 over urban, suburban and rural regions, and all correlate well with those of MEE data. Meanwhile,

664 surface $NO_2$ concentrations derived from LEO satellite observations also agree well with those of

POMINO-GEMS, except that POMINO-GEMS derived surface $NO_2$ concentrations are higher than
those of GOME-2 GDP 4.8 by about 40% – 60%. We conclude that validation with extensive MEE
measurements presents promising performance of POMINO-GEMS retrievals, especially the great
agreement of POMINO-GEMS $NO_2$ diurnal variation with MEE data over urban, suburban and rural
regions.
**3.5 Error estimates for POMINO-GEMS tropospheric $NO_2$ VCDs**
Total retrieval errors for POMINO-GEMS tropospheric $NO_2$ VCDs are derived from the
calculations of total SCDs, stratospheric SCDs, and tropospheric AMFs. Spatial and temporal averaging
across GEMS pixels can greatly reduce the random errors, but will not affect the systematic errors. Here,
we provide a preliminary estimate of POMINO-GEMS errors for the summertime retrieval discussed
above.
As described in Section 2, we calculate hourly total SCDs based on the original GEMS SCD data
and daily TROPOMI-guided corrections. According to the GEMS ATBD of $NO_2$ retrieval algorithm, the
SCD errors from the DOAS method are < 5.65% at high-$NO_2$ conditions ($NO_2$ VCD > $1 \times 10^{15}$ molec.
$cm^{-2}$) (Lee et al., 2020). The $NO_2$ SCD errors of TROPOMI are reported to be $0.5 – 0.6 \times 10^{15}$ molec.
$cm^{-2}$ (10% in a relative sense) (Van Geffen et al., 2022a). Given the assumption we made in adjusting
GEMS total SCDs to match TROPOMI values, we tentatively estimate the error in our corrected total
SCD data to be $0.5 – 0.7 \times 10^{15}$ molec. $cm^{-2}$ (10% in a relative sense) for most regions and $0.9 \times 10^{15}$
molec. $cm^{-2}$ (20% – 30%) at the edge of the northwestern GEMS FOV.
In constructing the stratospheric $NO_2$ SCDs, the stratospheric VCDs are taken from TROPOMI PAL
v2.3.1, scaled based on GEOS-CF v1 stratospheric $NO_2$ to account for diurnal variation, and then applied
with geometric AMFs. We assign a constant error of $0.2 \times 10^{15}$ molec. $cm^{-2}$ (5% – 10%) to our hourly
stratospheric SCDs, the same as the value for TROPOMI (Van Geffen et al., 2022a). Few studies have
assessed the accuracy of stratospheric $NO_2$ and its diurnal variation from GEOS-CF data (Knowland et
al., 2022b), but our comparison between GEOS-CF and TROPOMI shows great consistency (Section
2.1.5). As most of the errors in total SCDs are absorbed in the stratosphere-troposphere separation step
(Van Geffen et al., 2015), the errors in tropospheric SCDs should be 10% – 30% depending on different
cases, with higher relative biases in cleaner situations.
Tropospheric AMF calculations are the dominant error source for retrieved tropospheric $NO_2$ VCDs

over polluted regions. According to Liu et al. (2020), the AMF errors caused by uncertainty in surface reflectance are about 10%, and errors induced by uncertainties in aerosol parameters are about 10% in clean regions and 20% for heavily polluted situations. We further assume that the $O_2$-$O_2$ cloud retrieval algorithm introduces another error at the 10% level to the $NO_2$ AMFs. The uncertainty in a priori $NO_2$ vertical profiles is estimated to cause an AMF error by 10% (Liu et al., 2020). Yang et al. (2023) suggested that the $NO_2$ profiles from GEOS-Chem (version 13.3.4) might contain incorrect timing of PBL mixing growth in the morning and thus introduce a relative root-mean-square error of 7.6% and NMB of 2.7% in AMF; however, this error could be greatly dampened by averaging over a long time period. The free tropospheric $NO_2$ bias in GEOS-Chem $NO_2$ profiles might also contribute to the retrieval errors especially over remote regions. Adding these errors in quadrature leads to the overall AMF errors for POMINO-GEMS at 20% – 40%.

The overall uncertainty in POMINO-GEMS tropospheric $NO_2$ VCDs is estimated by adding in quadrature the errors in tropospheric $NO_2$ SCDs and AMFs, when these errors are expressed in the relative sense. For remote regions with low tropospheric $NO_2$ abundances, the overall retrieval uncertainties can reach 30% – 50% and are dominated by errors in tropospheric SCDs. For regions with abundant tropospheric $NO_2$, the uncertainties of retrieved tropospheric VCDs are dominated by the AMF errors and are estimated to be about 20% – 30%.

As shown in Figure 8d and Figure 11d, the maximum negative NMB of POMINO-GEMS tropospheric $NO_2$ VCDs relative to ground-based MAX-DOAS data is about 20% in the mid-morning, and the NMB of POMINO-GEMS derived surface $NO_2$ concentrations to MEE measurements is −30% on average. Thus our estimated error magnitude is supported by the independent ground-based MAX-DOAS and MEE data.

**4. Conclusions**

The GEMS instrument provides an unprecedented opportunity for air quality monitoring at a high spatiotemporal resolution. Our POMINO-GEMS algorithm retrieves tropospheric $NO_2$ VCDs as a research product. The algorithm first calculates hourly tropospheric $NO_2$ SCDs through fusion of total $NO_2$ SCDs from the GEMS v1.0 L2 $NO_2$ product, total and stratospheric $NO_2$ columns from the TROPOMI PAL v2.3.1 L2 $NO_2$ product, and stratospheric $NO_2$ diurnal variations from the GEOS-CF v1 dataset. The fusion approach reduces the high bias in total SCDs and removes the stripe-like patterns in

the official GEMS v1.0 product. Our algorithm then calculates tropospheric $NO_2$ AMFs to convert SCDs
to VCDs. A preliminary estimate of retrieval errors is also given.
Our initial POMINO-GEMS data for JJA 2021 shows high values of tropospheric $NO_2$ VCDs with
clear hotspots ($> 10 \times 10^{15}$ molec. cm$^{-2}$) over regions where anthropogenic emissions of $NO_x$ are abundant.
The spatial gradients of tropospheric $NO_2$ VCDs from urban centers to surrounding areas are substantial
in the morning due to traffic emissions, but the gradients are much reduced at noon and in the afternoon.
A gradual increase of tropospheric $NO_2$ VCDs from the morning to noon is observed over clean regions
of western China, likely as a result of enhanced biogenic emissions. Over high $NO_2$ regions where
anthropogenic activities dominate the $NO_x$ emissions, $NO_2$ columns increase until a peak at 09:00 – 10:00
LST, decrease to the minimum at noon and then increase in the afternoon again. Such characteristics of
$NO_2$ diurnal variations are associated with the changes in natural and anthropogenic $NO_x$ emissions,
photochemistry and atmospheric transport.
POMINO-GEMS tropospheric $NO_2$ VCDs agree well with POMINO-TROPOMI v1.2.2 in terms of
spatial correlation (0.98) and NMB (4.9%). POMINO-GEMS data are also consistent with the OMNO2
v4 tropospheric $NO_2$ VCD product in the early afternoon and GOME-2 GDP 4.8 tropospheric $NO_2$ VCD
product in the morning, with $R$ of 0.87 and 0.83, and NMB of $-16.8\%$ and $-1.5\%$, respectively.
POMINO-GEMS tropospheric $NO_2$ VCDs are comparable with ground-based MAX-DOAS
measurements at nine ground-based sites with a small NMB ($-11.1\%$), although the correlation is modest
($R = 0.66$). Both the bias and correlation values are smaller than POMINO-TROPOMI v1.2.2 (NMB =
$-18.1\%$, $R = 0.83$). More importantly, POMINO-GEMS well captures the diurnal variation of MAX-
DOAS $NO_2$ VCDs at Xuzhou ($R = 0.82$), Hefei ($R = 0.96$), Fudan University ($R = 0.84$), Nanhui ($R =
0.79$), Xianghe ($R = 0.94$) and Dianshan Lake ($R = 0.60$) sites, although the correlations are relatively
poor at Chongming and Fukue sites. Comparison with mobile-car MAX-DOAS measurements in the
Three Rivers' Source region on the Tibetan Plateau also shows good correlation in $NO_2$ diurnal variation
($R = 0.81$).
We also compare surface $NO_2$ concentrations derived from tropospheric $NO_2$ VCDs in POMINO-
GEMS and POMINO-TROPOMI v1.2.2 against MEE data, taking advantage of the large number of
MEE sites. POMINO-GEMS derived surface $NO_2$ concentration data exhibit a small NMB ($-26.3\%$).
For these sites at TROPOMI overpass times, POMINO-GEMS derived surface $NO_2$ concentrations show

a smaller magnitude of NMB (−48.0%) than POMINO-TROPOMI v1.2.2 (−55.8%). Excellent agreement in diurnal variation between POMINO-GEMS derived and MEE $NO_2$ is exhibited over all ($R$ = 0.97), urban ($R$ = 0.97), suburban ($R$ = 0.97) and rural ($R$ = 0.96) sites.

Overall, our comprehensive validation process highlights the good performance of POMINO-GEMS tropospheric $NO_2$ VCD product, both in magnitude and spatiotemporal variation. However, there are still several limitations in our study. To address the systematic overestimation and stripes problems in the original GEMS data, we correct GEMS total $NO_2$ SCDs by using TROPOMI data as a temporary solution. For example, we implement a simple geometric correction to combine GEMS and TROPOMI total $NO_2$ SCDs, but their differences in scattering geometry are only partly accounted for. Thus this correction works well in most regions, but may introduce SCD uncertainties up to $0.9 \times 10^{15}$ molec. $cm^{-2}$ (20% – 30%) at the edge of the northwestern GEMS FOV. Currently, the Environmental Satellite Center of South Korea is updating the $NO_2$ SCD data to v2.0. We will update our POMINO-GEMS algorithm accordingly, once the updated official $NO_2$ product becomes available to provide necessary inputs for our research product. In addition, in the conversion from $NO_2$ VCDs to surface concentrations, we use a constant correction factor of 2 to account for the strong $NO_2$ vertical gradient near the surface. This simple treatment does not account for the diurnal variation of the correction factor, and thus may introduce errors in the derived surface $NO_2$ concentrations. Nevertheless, the current POMINO-GEMS data serve as our initial attempt to derive the diurnal variations of tropospheric $NO_2$ at a high spatiotemporal resolution from GEMS, and they are expected to offer a useful source of information for various applications such as air quality analysis and emission constraint.

*Data availability*. The POMINO-GEMS $NO_2$ data will be freely available soon at the ACM group product website (http://www.pku-atmos-acm.org/acmProduct.php/). The TROPOMI PAL v2.3.1 L2 product can be downloaded from https://data-portal.s5p-pal.com. The OMNO2 v4 L2 product can be downloaded from https://aura.gesdisc.eosdis.nasa.gov/data/Aura_OMI_Level2/OMNO2.003/. The GOME-2 GDP 4.8 L2 product can be downloaded from http://acsaf.org/ after registration. The GEOS-CF v1.0 dataset can be downloaded from https://gmao.gsfc.nasa.gov/weather_prediction/GEOS-CF/data_access/. The MEE surface $NO_2$ measurements can be downloaded from https://quotsoft.net/air/. The ground-based and mobile-car MAX-DOAS measurements can be provided upon requests to the

corresponding owners.

*Author contributions*. JL conceived this research. YZ and JL designed the algorithm and validation
process. YZ performed all calculations with additional code support from HK. YZ and JL wrote the paper.
RS provided LIDORT. JK, HL, JP and HH provided GEMS data. MVR, FH, TiW, PW, QH, KQ, YC,
YK, JX, PX, XT, SZ and SW provided the ground-based MAX-DOAS measurements. SC, XC, JM and
ThW provided the mobile-car MAX-DOAS measurements. HK helped process MEE measurements. LC
and ML helped analyze the validation results. All authors commented on the paper.

*Competing interests*. The authors declare that they have no conflicts of interest.

*Financial support*. This research has been supported by the National Natural Science Foundation of
China (grant no. 42075175) and the Second Tibetan Plateau Scientific Expedition and Research Program
(grant no. 2019QZKK0604).

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
