# Peer review of "POMINO-GEMS: A Research Product for Tropospheric NO2 Columns from"

_Atmospheric Measurement Techniques, 2023_

## Author Comment (AC1)

**Authors' response to comments from Anonymous Referee #1**

**General comments:**

This paper presents $NO_2$ results from the GEMS instrument for June-August 2021 using the POMINO algorithm involves detailed improvements to cloud retrieval, surface reflectance and profiles and has previously been applied to measurements over Asia from TROPOMI and OMI. This is the first $NO_2$ retrieval I have seen from GEMS in the refereed literature, and it is exciting to see this first attempt at $NO_2$ retrievals.

Unfortunately, the early operational version of GEMS $NO_2$ slant columns retrieval have shown significant bias, and so the authors apply a scaling to the slant columns using TROPOMI and GEOS-Chem. This is not ideal, but at least allows the authors to proceed with a comprehensive $NO_2$ retrieval while the $NO_2$ slant columns retrievals are being improved.

The comparisons with MAX-DOAS and an extensive set of surface monitors show some biases but are actually pretty promising for a first attempt of $NO_2$ retrievals from geostationary orbit. It would be nice to see a more detailed discussion of possible uncertainties in the product, but on the other hand, this is a first attempt and there will likely be campaigns and retrieval improvements to come that will help to isolate error sources, and perhaps those discussions can be saved for future work.

Overall, I think this is a well-written and clear paper, and a careful first analysis of GEMS data. I recommend it be published after addressing a few minor comments.

We sincerely thank the Referee #1 for reviewing our paper and providing constructive comments for improvement. We updated our POMINO-GEMS algorithm by replacing nested GEOS-Chem v9-02 derived stratospheric $NO_2$ VCDs with NASA GEOS-CF v1 derived stratospheric $NO_2$ VCDs, and reprocessed all retrievals. Updated validation results show great improvement in $NO_2$ diurnal patterns between POMINO-GEMS and ground-based MAX-DOAS measurements. We also use mobile-car MAX-DOAS measurements in the Three Rivers' Source region on the Tibetan Plateau to validate POMINO-GEMS retrievals, and good agreement is also shown in terms of $NO_2$ diurnal variation. Responses to these general and specific comments are provided below.

As this is the first geostationary mission able to measure $NO_2$, it would be nice to see more discussion about sources of diurnal uncertainties (even qualitative discussion). The MAX-DOAS and GEMS $NO_2$ seem to have different trends in the afternoon measurements at many sites. What could cause this?

Validation results of updated POMINO-GEMS tropospheric $NO_2$ VCDs using ground-based MAX-DOAS measurements show much improved and great correlation of $NO_2$ diurnal variations at Xuzhou ($R = 0.82$), Hefei ($R = 0.96$), Fudan University ($R = 0.84$), Nanhui ($R = 0.79$), Xianghe ($R = 0.94$) and Dianshan Lake ($R = 0.60$) sites, even though the correlations are modest at Chongming (rural) and Fukue (remote) sites. We have added more discussion in the revised manuscript.
Currently, we are not able to quantitatively attribute the sources of POMINO-GEMS tropospheric $NO_2$ diurnal uncertainties for each hour and pixel. However, in a qualitative perspective, the retrieval

uncertainties for remote regions might vary little during the daytime, which are likely caused by choices on the reference spectra used in spectral fitting processes. Over polluted regions, the diurnal uncertainties of aerosol extinction profiles and a priori $NO_2$ profiles are likely to be the dominant sources for the diurnal $NO_2$ retrieval uncertainties. We will quantify the sources of diurnal retrieval uncertainties in future studies.

In Line 543-559, we added:

"Figure 9 compares the diurnal variation of tropospheric $NO_2$ VCDs between POMINO-GEMS and MAX-DOAS at eight stations. At each site, $NO_2$ values are averaged in JJA 2021 at each hour for comparison, and the number of valid days for each hour is also shown. The Cape Hedo site is not included because there are few valid MAX-DOAS data points at each hour. Figure 10a-f show that at the urban and suburban sites, MAX-DOAS $NO_2$ (black lines) peaks in the mid-to-late morning, declines towards the minimum values at noon around 13:00 LST, and then gradually increases in the afternoon. Strong correlation of $NO_2$ diurnal variation between POMINO-GEMS (red solid lines) and MAX-DOAS is found at Xuzhou ($R$ = 0.82), Hefei ($R$ = 0.96), Fudan University ($R$ = 0.84), Nanhui ($R$ = 0.79) and Xianghe ($R$ = 0.94). At the Dianshan Lake site, POMINO-GEMS $NO_2$ columns increase but MAX-DOAS data decrease from 08:00 to 09:00 LST, resulting in a lower correlation coefficient ($R$ = 0.60). At Chongming and Fukue sites, MAX-DOAS $NO_2$ shows a peak in the morning without evident increase in the early afternoon, but this diurnal pattern is not fully captured by POMINO-GEMS. At Fukue, POMINO-GEMS $NO_2$ exhibit abrupt changes at 12:00 and 13:00 LST due to few valid data.

In addition, comparison of POMINO-GEMS diurnal variation with $NO_2$ data from GOME-2 in the morning and OMI and TROPOMI in the early afternoon shows good agreement at Hefei, Nanhui, Dianshan Lake, Chongming and Fukue sites. The differences between POMINO-GEMS to MAX-DOAS $NO_2$ VCDs are comparable or smaller than those between LEO satellite and MAX-DOAS $NO_2$ VCDs."

How accurate are the GEOS-Chem profiles over a day? Do they look like the MAX-DOAS profiles?

We agree that it is important to know the performance of GEOS-Chem on simulating $NO_2$ vertical profiles. Although we didn't evaluate the accuracy and diurnal variations of nested GEOS-Chem v9-02 simulated $NO_2$ profiles, they have been used in our POMINO-OMI and POMINO-TROPOMI research products. Validation results show higher accuracy of previous POMINO products compared with independent ground-based measurements (Liu et al., 2019; Liu et al., 2020). Besides, Yang et al. (2023) have tested the performance of GEOS-Chem CTM to simulate hourly $NO_2$ vertical profiles for GEMS AMF calculation. Therefore, we quoted the discussion of Yang et al. (2023) in Section 3.5 to briefly discuss the AMF uncertainties from $NO_2$ profiles. In addition, we do not have ground-based MAX-DOAS $NO_2$ profiles at any station, so we cannot compare and discuss the $NO_2$ profiles between GEOS-Chem simulations and MAX-DOAS measurements.

In Line 697-703, we added:
"The uncertainty in a priori $NO_2$ vertical profiles is estimated to cause an AMF error by 10% (Liu et al., 2020). Yang et al. (2023) suggested that the $NO_2$ profiles from GEOS-Chem (version 13.3.4) might contain incorrect timing of PBL mixing growth in the morning and thus introduce a relative root-mean-square error of 7.6% and NMB of 2.7% in AMF; however, this error could be greatly dampened by averaging over a long time period. The free tropospheric $NO_2$ bias in GEOS-Chem $NO_2$ profiles might

also contribute to the retrieval errors especially over remote regions."

Are any errors expected from the application of a LEO BRDF to an AMF calculation?

It is true that systematic errors might arise by using LEO BRDF to calculate AMFs for GEMS. However, we think these errors are much less than those from aerosol corrections and priori $NO_2$ profiles. Since GEMS BRDF L2 product has been available, we will test the differences caused by this issue in the future, and replace current MODIS BRDF data with GEMS product if necessary.

Also, there is no discussion of MAX-DOAS uncertainties themselves.

We have added discussion of MAX-DOAS uncertainties themselves in the revised manuscript.

In Line 347-349, we added:
"Kanaya et al. (2014) and Hendrick et al. (2014) have discussed the error in MAX-DOAS $NO_2$ retrieval: uncertainties from a priori aerosol and $NO_2$ profiles are the largest source by 10% – 14%, and the total retrieval uncertainty is typically 12% – 17%."

**Specific comments:**

Line 75: This is a specific technique that is used for many missions and trace gases, but not all (for instance, direct fitting of radiances can also be used). Suggest change to more general "using spectral fitting" or similar.

Thank you for your suggestion. We have changed the expression to "The first step is to retrieve total $NO_2$ slant column densities (SCDs) with spectral fitting techniques, such as the Differential Optical Absorption Spectroscopy (DOAS)".

Line 88: Are they using an online calculation or look up tables based on VLIDORT?

They use a precomputed look-up table of box AMFs based on VLIDORT version 2.6. We have updated the description in the revised manuscript (Line 97-99).

Line 154: "daily $NO_2$, pressure, temperature and aerosol vertical profiles". These haven't been introduced yet. Are they coming from the GEOS-Chem model or TROPOMI?

They come from nested GEOS-Chem v9-02 simulations. We have updated the description in the revised manuscript (Line 167-169).

Figure 2 caption: Is GEMS product only at TROPOMI overpass time or all hours? Described in text but should also be mentioned in caption.

Thank you for your suggestion. We have updated Figure 2 in the revised manuscript.

Section 2.1.3: there must be several assumptions made to use this method of scaling GEMS to TROPOMI. Can you mention them? For instance, geometric AMFs won't account for GEO vs LEO issues like relative azimuth angle. Do these make any difference?

Thank you for your suggestion. We have added a paragraph to discuss the assumption we make in this correction.

In Line 237-241, we added:
"In Eq. (2), we implement a simple geometric correction (concerning SZAs and VZAs) for AMFs instead of using the actual AMFs; the latter could account for the differences in relative azimuth angles and other factors. Specific derivation of this assumption is given in Section 1 of the Supplement Information (SI). The correction is assumed to be acceptable with an extra uncertainty introduced to the total $NO_2$ SCDs, as will be further discussed in Section 3.5."

Line 232: What do you use over water where BRDF is not available (open ocean) or is inaccurate (for example in coastal regions)?

We use MODIS BRDF coefficients over land and coastal ocean regions, and OMLER v3 albedo over open ocean. We have updated the sentences in the revised manuscript (Line 306-309).

Line 262: Since these data are being used for validation, it would be good to further justify "multiplied by a factor of 2 to roughly account". Does $NO_2$ necessarily change linearly in those bottom 130 m?

We use the constant correction factor of 2 based on Liu et al. (2018). Figure 12 in this paper (shown below) compares mean $NO_2$ vertical profiles over Eastern China from nested GEOS-Chem v9-02 and WRF/CMAQ v5.0.1. CMAQ simulations show much stronger vertical gradient of $NO_2$ from its first layer (about 40 m) to its second layer (about 80 m), but GEOS-Chem cannot fully capture the vertical gradient of $NO_2$ concentrations. Therefore, we decided to roughly account for this issue by implementing a simple correction with a factor of 2. We admit that $NO_2$ concentrations don't necessarily change linearly below 130 m, so the correction factor of 2 must introduces systematic bias of satellite derived surface $NO_2$ concentrations. However, the diurnal variations of satellite derived surface $NO_2$ concentrations are still consistent with those of MEE data, with correlation coefficients great than 0.96. We have added more sentences on this issue in the revised manuscript.

[Figure]

**Figure 12.** Eastern China mean $NO_2$ vertical profiles simulated by GEOS-Chem and CMAQ averaged over 25 October–25 December 2013. The black and red dots denote the center of each vertical layer in the two models. The evening is from 20:00 to 23:00 LT, while the afternoon is from 12:00 to 15:00 LT.

In Line 339-341, we added:

"However, the constant correction factor of 2 neglects the diurnal variation of $NO_2$ vertical gradient, which is related to the diurnal variation of planetary boundary layer (PBL) heights. This issue is discussed in detail in Section 3.4"

In Line 631-639, we added:

"The discrepancies between POMINO-GEMS and MEE surface $NO_2$ concentrations at different hours are likely caused by the assumed constant correction factor of 2 to account for the vertical gradient of $NO_2$ from the height of ground instrument to the center of the first model layer (Section 2.2). In the morning when the PBL is low, most $NO_2$ molecules are near the ground and the vertical gradient of $NO_2$ over polluted regions is the largest in the daytime, so the factor of 2 may lead to underestimation of derived surface $NO_2$ concentrations. In contrast, in the afternoon, the PBL mixing is much stronger and the vertical gradient of $NO_2$ is much smaller, thus the factor of 2 may lead to overestimated surface $NO_2$ concentrations."

Section 2.4: What are uncertainties in MEE measurements and what are the details of the observations? Are they chemiluminescence measurements that suffer from bias in $NO_2$? This is mentioned later but I think is appropriate to include in this section.

Thank you for your suggestion. We have updated the description of the details of MEE measurements in Section 2.5.

In Line 396-402, we added:

"At MEE sites, molybdenum catalyzed conversion from $NO_2$ to NO and subsequent chemiluminescence measurement of NO is done to estimate $NO_2$ concentrations. The heated molybdenum catalyst has low

chemical selectivity, leading to strong interference from other oxidized nitrogen species such as nitric acid ($HNO_3$) and peroxyacetyl nitrate (PAN). Therefore, MEE data tend to overestimate the actual $NO_2$ concentrations, with the extent of overestimation about 10% – 50% (Boersma et al., 2009; Liu et al., 2018). The overestimation is dependent on the oxidation level of $NO_x$, but is currently unclear for each site and hour."

Figure 7 and Line 342-354: The bias between GEMS and TROPOMI is different between ocean and land. Several reasons are given but I don't understand why these would product different bias over land and water – is it just that the bias are actually following locations of no aerosols vs. high aerosols and not necessarily associated with water/land? Is there any way that the surface itself can influence this bias?

The writing here is indeed misleading. We have added a section (Section 3) in the Supplement Information (SI) to discuss the reasons for the differences between POMINO-GEMS and POMINO-TROPOMI v1.2.2 tropospheric $NO_2$ VCDs. Besides, previous studies have shown the effects of surface reflectance on $NO_2$ retrieval, but there is no apparent relationship between bias and surface conditions (Zhou et al., 2010; Lin et al., 2015; Vasilkov et al., 2016).

Line 455: "assume no error contributions from the GEOS-Chem-based scaling": Wondering here on what this assumption is based? Are there any references describing accuracy of diurnal variation of $NO_2$ from GEOS-Chem?

Since we have decided to replace GEOS-Chem stratospheric $NO_2$ VCDs with those from GEOS-CF, we have updated the sentences in Section 3.5. As far as we know, there is no study validating the diurnal variations of stratospheric $NO_2$ from GEOS-CF v1 product, but our comparison between GEOS-CF and TROPOMI shows great consistency. Therefore, the GEOS-CF v1 dataset is in general reliable in our algorithm.

In Line 684-690, we added:
"In constructing the stratospheric $NO_2$ SCDs, the stratospheric VCDs are taken from TROPOMI PAL v2.3.1, scaled based on GEOS-CF v1 stratospheric $NO_2$ to account for diurnal variation, and then applied with geometric AMFs. We assign a constant error of $0.2 \times 10^{15}$ molec. $cm^{-2}$ (5% – 10%) to our hourly stratospheric SCDs, the same as the value for TROPOMI (Van Geffen et al., 2022). Few studies have assessed the accuracy of stratospheric $NO_2$ and its diurnal variation from GEOS-CF data (Knowland et al., 2022), but our comparison between GEOS-CF and TROPOMI shows great consistency (Section 2.1.5)."

Line 460: Related to previous comment, how good are $NO_2$ a priori profiles from the model at various times of day? Does uncertainty vary over the days? Also, there is a free troposphere $NO_2$ bias in GEOS-Chem which can give large errors in $NO_2$ measurements over remote regions – maybe mention this as a source of uncertainty.

We have quoted the discussion of Yang et al. (2023) which tested the ability of GEOS-Chem CTM to simulate hourly $NO_2$ vertical profiles for GEMS AMF calculation. The uncertainty of $NO_2$ a priori profiles is largest in the morning due to incorrect model timing of PBL mixing growth, but becomes

much smaller in the afternoon.

Thank you for your suggestion about the free troposphere $NO_2$ bias in GEOS-Chem. We have added it in our error analysis.

In Line 698-703, we added:

"The uncertainty in a priori $NO_2$ vertical profiles is estimated to cause an AMF error by 10% (Liu et al., 2020). Yang et al. (2023) suggested that the $NO_2$ profiles from GEOS-Chem (version 13.3.4) might contain incorrect timing of PBL mixing growth in the morning and thus introduce a relative root-mean-square error of 7.6% and NMB of 2.7% in AMF; however, this error could be greatly dampened by averaging over a long time period. The free tropospheric $NO_2$ bias in GEOS-Chem $NO_2$ profiles might also contribute to the retrieval errors especially over remote regions."

**Technical comments:**

Please define POMINO acronym early on. I'm not sure what it stands for.

Done.

Figure 6: Consider adding another set of lat/lon values on the axes. For someone not very familiar with the shape of Chinese provinces, it's hard to figure out the region being examined.

Done.

Figure S5: I find this figure very hard to read, even when zooming. Perhaps increasing the resolution would help (or maybe color palette and/or symbol size?). The sub-figures are even harder to decipher. What are these – they are lacking circles and it's not clear if they are a measurement like the others?

Updated.

Line 258: write out "molecules" instead of using "molec".

Done.

[revised manuscript text omitted]

---

## Author Comment (AC2)

**Authors' response to comments from Anonymous Referee #2**

**General comments:**

This paper presents $NO_2$ results from the GEMS instrument for June-August 2021. As the $NO_2$ slant columns are biased, the authors present a correction at S5P overpass time based on the TROPOMI $NO_2$ SCD. The stratospheric correction is also based on the TROPOMI $NO_2$ product, and on the GEOS-Chem model, including its stratospheric diurnal variation. the POMINO algorithm is then applied to derive the AMF and the final $NO_2$ tropospheric columns. The POMINO-GEMS $NO_2$ columns are finally compared with the POMINO-TROPOMI product, as well as MAX-DOAS columns and $NO_2$ surface concentrations.

The paper is well-written and clear. I recommend publication after addressing the above major comments.

We thank the Referee #2 for taking time to review our paper and provide constructive suggestions and comments for improvement. We updated our POMINO-GEMS algorithm by replacing nested GEOS-Chem v9-02 derived stratospheric $NO_2$ VCDs with NASA GEOS-CF v1 derived stratospheric $NO_2$ VCDs, and reprocessed all retrievals. Updated validation results show great improvement in $NO_2$ diurnal patterns between POMINO-GEMS and ground-based MAX-DOAS measurements. We also use mobile-car MAX-DOAS measurements in the Three Rivers' Source region on the Tibetan Plateau to validate POMINO-GEMS retrievals, and good agreement is also shown in terms of $NO_2$ diurnal variation. Responses to these general and specific comments are provided below.

My main concern is the strong correction applied to the GEMS observations. At S5P overpass time, the GEMS $NO_2$ SCD are basically replaced by the TROPOMI $NO_2$ SCD, on a grid cell basis. To my understanding, the only true GEMS $NO_2$ information remaining is the diurnal variation relative to the mid-morning values. Unfortunately, the MAX-DOAS validation results are poor when it comes to diurnal variations. This is a serious limitation. This should be further discussed in the paper.

Thank you very much for your suggestion. We found that the poor correlations of $NO_2$ diurnal variations between POMINO-GEMS and ground-based MAX-DOAS measurements are mainly caused by poor simulation of stratospheric $NO_2$ from nested GEOS-Chem v9-02. Therefore, we decided to use NASA GEOS-CF v1 product to re-calculate hourly stratospheric $NO_2$ VCDs. Updated comparison results with ground-based MAX-DOAS measurements show much better correlations in terms of $NO_2$ diurnal variation, and we also proved that TROPOMI-guided correction for total $NO_2$ SCDs makes little difference to the POMINO-GEMS $NO_2$ diurnal variations. We have added more discussion about this issue in the revised manuscript.

In Line 543-577, we added:
    "Figure 9 compares the diurnal variation of tropospheric $NO_2$ VCDs between POMINO-GEMS and MAX-DOAS at eight stations. At each site, $NO_2$ values are averaged in JJA 2021 at each hour for comparison, and the number of valid days for each hour is also shown. The Cape Hedo site is not included because there are few valid MAX-DOAS data points at each hour. Figure 10a-f show that at the urban and suburban sites, MAX-DOAS $NO_2$ (black lines) peaks in the mid-to-late morning, declines towards the minimum values at noon around 13:00 LST, and then gradually increases in the afternoon. Strong

correlation of NO$_2$ diurnal variation between POMINO-GEMS (red solid lines) and MAX-DOAS is found at Xuzhou ($R$ = 0.82), Hefei ($R$ = 0.96), Fudan University ($R$ = 0.84), Nanhui ($R$ = 0.79) and Xianghe ($R$ = 0.94). At the Dianshan Lake site, POMINO-GEMS NO$_2$ columns increase but MAX-DOAS data decrease from 08:00 to 09:00 LST, resulting in a lower correlation coefficient ($R$ = 0.60). At Chongming and Fukue sites, MAX-DOAS NO$_2$ shows a peak in the morning without evident increase in the early afternoon, but this diurnal pattern is not fully captured by POMINO-GEMS. At Fukue, POMINO-GEMS NO$_2$ exhibit abrupt changes at 12:00 and 13:00 LST due to few valid data.

In addition, comparison of POMINO-GEMS diurnal variation with NO$_2$ data from GOME-2 in the morning and OMI and TROPOMI in the early afternoon shows good agreement at Hefei, Nanhui, Dianshan Lake, Chongming and Fukue sites. The differences between POMINO-GEMS to MAX-DOAS NO$_2$ VCDs are comparable or smaller than those between LEO satellite and MAX-DOAS NO$_2$ VCDs.

As we use TROPOMI total NO$_2$ SCDs to correct those of GEMS, this may influence the NO$_2$ diurnal variation of original GEMS observations. Thus we also compare MAX-DOAS data with re-calculated POMINO-GEMS tropospheric NO$_2$ VCDs without correction in total SCDs (red dashed lines in Figure 9). Compared to our default POMINO-GEMS data (with correction), excluding the correction leads to lower diurnal correlation coefficients at Xuzhou, Hefei, Fudan University, Nanhui and Dianshan Lake, but higher correlation coefficients at Xianghe, Chongming and Fukue. Excluding the correction increases the NMB at three sites but decreases the NMB at five sites. We conclude that at these eight sites (in the eastern areas), no significant influences on the diurnal variation of POMINO-GEMS tropospheric NO$_2$ VCDs are brought in through TROPOMI-based correction for total NO$_2$ SCDs."

On the same idea, the authors present a comparison between POMINO-GEMS and POMINO-TROPOMI. The comparison results are obviously very good, but the study is biased. I strongly recommend to use independent satellite NO$_2$ products; such as OMI and GOME-2 products. The addition of OMI and GOME-2 would allow to compare with the GEMS observed diurnal variation.

Thank you for your constructive suggestion. We have added comparisons between POMINO-GEMS and OMNO2 v4 and GOME-2 GDP 4.8 tropospheric NO$_2$ products, and made additional comparisons in the discussion for NO$_2$ diurnal variations.

In Line 496-506, we added:
"Figure 7d-f and g-i show the comparison results of POMINO-GEMS tropospheric NO$_2$ VCDs with OMNO2 v4 on a 0.25° × 0.25 °grid and GOME-2 GDP 4.8 on a 0.5 °× 0.5 °grid averaged over JJA 2021, respectively. POMINO-GEMS NO$_2$ VCDs exhibit good spatial consistency with the two independent products ($R$ = 0.87 and 0.83), although with slightly lower values than OMNO2 v4 (by 16.8%) and GOME-2 GDP 4.8 (by 1.5%). These VCD differences are expected, considering the differences in the retrieval algorithm. For example, the POMINO-GEMS algorithm implements explicit aerosol corrections in the radiative transfer calculation, while OMNO2 v4 and GOME-2 GDP 4.8 treat aerosols as "effective clouds". POMINO-GEMS accounts for the anisotropy of surface reflectance by adopting MODIS BRDF coefficients, whereas OMNO2 v4 and GOME-2 GDP 4.8 use geometry-dependent and regular LER, respectively. The horizontal resolution of a priori NO$_2$ profiles in POMINO-GEMS is 25 km (and interpolated to 2.5 km), 1° × 1.25 °in OMNO2 v4 and 1.875 °× 1.875 °in GOME-2 GDP 4.8."

In Line 556-559, we added:

"In addition, comparison of POMINO-GEMS diurnal variation with $NO_2$ data from GOME-2 in the morning and OMI and TROPOMI in the early afternoon shows good agreement at Hefei, Nanhui, Dianshan Lake, Chongming and Fukue sites. The differences between POMINO-GEMS to MAX-DOAS $NO_2$ VCDs are comparable or smaller than those between LEO satellite and MAX-DOAS $NO_2$ VCDs"

In Line 663-666, we added:
"Meanwhile, surface $NO_2$ concentrations derived from LEO satellite observations also agree well with those of POMINO-GEMS, except that POMINO-GEMS derived surface $NO_2$ concentrations are higher than those of GOME-2 GDP 4.8 by about 40% – 60%."

Since this is the first study about GEMS $NO_2$ measurements, the paper should provide a section where the GEMS operational VCDs are compared to the presented product and provide some conclusions on the regions and periods where the GEMS $NO_2$ tropospheric VCD are performing good or bad.

Thank you for your suggestion. We agree that comparison between POMINO-GEMS and GEMS operational $NO_2$ product is necessary. Unfortunately, we found that tropospheric $NO_2$ VCDs in GEMS v1 operational product in summer are unavailable (no valid data), so we couldn't perform the comparison. As soon as the reprocessing of GEMS v2.0 operational product is finished, we will compare the updated GEMS operational tropospheric $NO_2$ VCDs with POMINO-GEMS retrievals.

In the diurnal variation plot (figure 9 and figure 11), the uncorrected GEMS $NO_2$ VCD should also be plotted. (uncorrected GEMS $NO_2$ VCD = uncorrected GEMS $NO_2$ SCD – $NO_2$ stratospheric columns)/POMINO GEMS AMFs.

Thank you for your suggestion. We have added the diurnal variation of uncorrected POMINO-GEMS tropospheric $NO_2$ VCDs in Figure 9 and Figure 10. The comparison results of corrected and uncorrected POMINO-GEMS derived surface $NO_2$ concentrations against MEE data are very similar, so we listed the statistics in Table S4 of the Supplement Information (SI).

In Line 569-577, we added:
"As we use TROPOMI total $NO_2$ SCDs to correct those of GEMS, this may influence the $NO_2$ diurnal variation of original GEMS observations. Thus we also compare MAX-DOAS data with re-calculated POMINO-GEMS tropospheric $NO_2$ VCDs without correction in total SCDs (red dashed lines in Figure 9). Compared to our default POMINO-GEMS data (with correction), excluding the correction leads to lower diurnal correlation coefficients at Xuzhou, Hefei, Fudan University, Nanhui and Dianshan Lake, but higher correlation coefficients at Xianghe, Chongming and Fukue. Excluding the correction increases the NMB at three sites but decreases the NMB at five sites. We conclude that at these eight sites (in the eastern areas), no significant influences on the diurnal variation of POMINO-GEMS tropospheric $NO_2$ VCDs are brought in through TROPOMI-based correction for total $NO_2$ SCDs."

In Line 586-588, we added:
"In contrast, POMINO-GEMS without total SCD correction exhibits much poorer correlation with mobile-car MAX-DOAS data, due to the erroneous increase in the afternoon."

In line 639-640, we added:

"Note that the consistency between POMINO-GEMS and MEE data does not depend on the total SCD correction (Table S4)."

**Specific comments:**

Abstract

Line 33: I suggest to remove the very first sentence, that sounds a bit obvious and is already in the introduction: Nitrogen dioxide ($NO_2$) is a major air pollutant.

Done.

Line 35: LEO $NO_2$ retrievals are not limited only by insufficient temporal sampling, but also by retrieval uncertainties and spatial resolution. The two limitations exist also for GEMS.

Revised.

Line 37: at an unprecedented hourly resolution during the daytime.

Revised.

Line 41: "We then derive tropospheric $NO_2$ air mass factors (AMFs) with explicit corrections for the anisotropy of surface reflectance and aerosol optical effects, through pixel-by-pixel radiative transfer calculations." The authors do not present the impact of those two corrections in the rest of the paper. It should be either be presented in the manuscript (see my AMF comments later) or removed from the abstract.

Thank you for your suggestion. In this study, we didn't perform sensitivity tests to discuss the impacts of surface reflectance and aerosol optical effects, but we compare these ancillary parameters when comparing POMINO-GEMS with other satellite products. Therefore, we decided to keep the sentence in the abstract.

In line 500-506, we added:
"These VCD differences are expected, considering the differences in the retrieval algorithm. For example, the POMINO-GEMS algorithm implements explicit aerosol corrections in the radiative transfer calculation, while OMNO2 v4 and GOME-2 GDP 4.8 treat aerosols as "effective clouds". POMINO-GEMS accounts for the anisotropy of surface reflectance by adopting MODIS BRDF coefficients, whereas OMNO2 v4 and GOME-2 GDP 4.8 uses geometry-dependent and regular LER, respectively. The horizontal resolution of a priori $NO_2$ profiles in POMINO-GEMS is 25 km (and interpolated to 2.5 km), $1° \times 1.25°$ in OMNO2 v4 and $1.875° \times 1.875°$ in GOME-2 GDP 4.8."

Line 44: The term "reveals" is overused, since the $NO_2$ hotspot signals are well known from LEO observations.

Revised.

Line 45: As intended by the presented method, POMINO-GEMS $NO_2$ VCDs agree well with POMINO-TROPOMI v1.2.2 product. Please indicate in the abstract that the remaining differences are coming from AMF differences.

Revised.

Introduction

Line 66: the provided references are for $NO_2$ datasets rather than LEO mission themselves. Please add more appropriate references for GOME, OMI, GOME-2, TROPOMI.

Done.

Line 90: Validation results have shown the overall capability of the official GEMS $NO_2$ algorithm. I'm not sure this is true. You should provide reference to support this affirmation.

Done.

Method and data

Line 128: Please explain briefly what is meant by "continuum reflectances".

Done.

Line 140: Please explain briefly what is meant by "area-weighted oversampling technique".

Done.

Please provide basic information on the slant columns retrieval settings for GEMS and TROPOMI operational products: wavelength interval, cross-sections, reference spectrum.

Done.

Total $NO_2$ SCDs

The correction based on the TROPOMI SCDs is somehow radical, since it is calculated for every grid cell. Have you tested more softer corrections, for example based on much larger grid cells, or based on meridionally averaged grids?

Thank you for your comment. We have tested three different softer corrections, and the results are shown below, respectively.

Correction based on 20°×20 °averaged grid cells:

[Figure]

Correction based on meridionally averaged grid cells:

[Figure]

Correction based on zonal averaged grid cells:

[Figure]

(a) TROPOMI PAL v2.3.1     (b) GEMS v1.0 (based on TROPOMI)

(c) Zonally averaged grid cells (based on TROPOMI)     (d) Zonally averaged grid cells (02:45-07:45 UTC)

$\times 10^{15}$ molec. cm$^{-2}$

These correction methods can reduce the high bias over northern and northwestern GEMS FOV to various extents, but are not capable to remove stripes. Therefore, we think the correction method applied in our algorithm is effective enough to address those systematic issues in official GEMS product.

In Line 252-258, we added:

"Our correction method is done for each grid cell. We tested other correction methods by applying the same correction value to grid cells within a 20° × 20° domain, at the same latitude, or at the same longitude. These alternative methods can reduce the high bias over the northern and northwestern GEMS FOV to various extents, but cannot remove the stripes (not shown). We also note that our simple correction is a temporary solution before the aforementioned systematic problems in the official GEMS SCD retrieval are solved by improving spectral fitting. In Sections 3.3 and 3.4, we compare the diurnal variations of tropospheric NO$_2$ VCDs based on corrected and uncorrected GEMS SCDs."

More examples of figure 2b and d could be shown for other GEMS hours (maybe in the supplement).

Done.

Line 199: Please comment on the diurnal variation of the GEMS systematic problems. For example, is the high bias over northern and northwestern part of GEMS FOV constant during the day or does it increase?

Thank you for your suggestion. Comparisons of two products at different hours are shown in Figure S1 of the SI. The stripes remain significant at all hours, which is expected because this problem has nothing to do with the observation time. However, since GEMS observations are spatiotemporally matched with

those of TROPOMI, there is no direct comparison over the northwestern GEMS FOV from mid-morning to noon, so the diurnal variation of systematic high bias of GEMS total $NO_2$ SCDs cannot be clearly depicted and hence discussed yet.

AMFs

A figure presenting the POMINO GEMS amfs should be added, as well as a comparison with the POMINO TROPOMI AMFs.

Done.

Estimation of surface $NO_2$ concentrations

Please specify if the Rgc GEOS-Chem simulated ration is time dependent or constant. In other words, is there a diurnal variation of the model introduced with this correction? If yes, what is the observed GEMS diurnal variations if you use a constant ratio?

Thank you for your suggestion. The GEOS-Chem simulated column-to-surface ratio is time independent, so there is a diurnal variation of the model introduced with this correction. We have added the discussion of the GEMS $NO_2$ diurnal variations using a daily ratio.

In Line 641-647, we added:
"To quantify the influences of the diurnal variation of hourly column-to-surface ratio from GEOS-Chem simulations, we compare the MEE measurements with POMINO-GEMS derived surface $NO_2$ concentrations using daily column-to-surface ratio (Figure S15). As expected, POMINO-GEMS derived $NO_2$ concentrations show a similar diurnal variation as the tropospheric $NO_2$ VCDs do, with two peaks in the mid-morning and afternoon, and a minimum at noon. The temporal correlation coefficient with MEE is only about 0.23. Thus it is more reasonable to use hourly ratio for comparison with MEE measurements, as done in our study."

Line 277: Please explain briefly what is the grubbs statistical test and provide a reference.

Done.

Results and discussion

As the $NO_2$ total and stratospheric SCDs are almost the same by definition of the presented "fusion" technique between GEMS and TROPOMI, I suggest to skip section 3.2 and to replace it by a comparison of AMFs from POMINO GEMS and TROPOMI.

Thank you for your suggestion. Even though the fusion method leads to very similar $NO_2$ total SCDs and stratospheric VCDs, there are still slight differences in tropospheric $NO_2$ SCDs which is caused by different geometries between GEMS and TROPOMI. Therefore, we have added more detailed discussion of the reasons for the differences between POMINO-GEMS and POMINO-TROPOMI v1.2.2

tropospheric NO$_2$ VCDs in Section 3 of the SI.

Figure 8: the regression line values are exactly the same between plots a and b. this seems strange, please check.

Thank you for your comment. The updated regression results are shown in the revised manuscript.

Figure 9: please use a fixed scale, or at least only two different scales for high and background NO$_2$ levels.

Done.

Figure 11: I suggest to detail the comparison with MEE diurnal variations for different groups of sites (urban, rural, northeast, southwest China). This could provide more information on the regions where the GEMS diurnal variation is valid or not.

Thank you for your constructive suggestion. We have added detailed comparison with different groups of MEE sites in the revised manuscript.

In Line 389-395, we added:
"The spatial distribution of all MEE sites in the GEMS FOV is shown in Figure S8a, and that of MEE sites over urban, suburban and rural regions are shown in Figure S8b–d, respectively. The classification of sites is based on Tencent user location data with a horizontal resolution of 0.05° × 0.05 °for every 0.5 second from 31 August to 30 September 2021 (Figure S8e), adopted from previous work (Kong et al., 2022). Here, urban MEE sites are defined as where the mean location request times is larger than 50 times per second, suburban sites refer to 5-50 times per second, and rural sites refer to less than 5 times per second. The number of sites for urban, suburban and rural sites are 808, 554 and 71, respectively."

In Line 661-669, we added:
"Figure 12b-d show the comparison of NO$_2$ diurnal variations for different groups of MEE sites. The diurnal variations of POMINO-GEMS derived surface NO$_2$ concentrations show similar characteristics over urban, suburban and rural regions, and all correlate well with those of MEE data. Meanwhile, surface NO$_2$ concentrations derived from LEO satellite observations also agree well with those of POMINO-GEMS, except that POMINO-GEMS derived surface NO$_2$ concentrations are higher than those of GOME-2 GDP 4.8 by about 40% – 60%. We conclude that validation with extensive MEE measurements presents promising performance of POMINO-GEMS retrievals, especially the great agreement of POMINO-GEMS NO$_2$ diurnal variation with MEE data over urban, suburban and rural regions."

Since the uncertainties on the measured diurnal variations appear to be large, I suggest to applied to the MAX-DOAS measurements a similar "column to surface column transformation" as for the satellite columns, and to compare directly MEE and MAX-DOAS diurnal variations.

Thank you for your suggestion. We have added discussion about this comparison.

In Line 648-652, we added:

"To further test the reliability of our VCD-to-surface-concentration conversion method (Eq. (9)), we apply the same method to MAX-DOAS $NO_2$ VCDs and compare the resulting surface $NO_2$ concentrations with MEE data. As shown in Figure S16, the diurnal variation of MAX-DOAS derived surface $NO_2$ concentrations correlates well with that of MEE measurements ($R = 0.96$), in support of our conversion method."

Error estimates

10% error on the GEMS $NO_2$ SCD (or we cloud say on the TROPOMI $NO_2$ SCDs) seems to be underestimated. Furthermore, the diurnal variations of the error on the GEMS fits is not taken into account.

Thank you for your comment. We have re-written Section 3.5 to discuss the error estimates in a more detailed way. Although we are not able to assess the diurnal variations of the error on the GEMS fit alone, we have added a quantitative discussion about the diurnal variation of spatiotemporal correlation coefficients and NMBs of POMINO-GEMS to ground-based MAX-DOAS and MEE measurements in Section 3.3 and 3.4. We will do the detailed error analysis in the future.

Conclusions

The observed added-value of GEMS should be discussed in a more balanced way in the conclusions, as well as the current limitations.

Thank you for your comment. We have added the discussion of current limitations in this study.

In Line 756-768, we added:
"However, there are still several limitations in our study. To address the systematic overestimation and stripes problems in the original GEMS data, we correct GEMS total $NO_2$ SCDs by using TROPOMI data as a temporary solution. For example, we implement a simple geometric correction to combine GEMS and TROPOMI total $NO_2$ SCDs, but their differences in scattering geometry are only partly accounted for. Thus this correction works well in most regions but may introduce uncertainties up to 30% in the northwestern GEMS FOV. Currently, the Environmental Satellite Center of South Korea is updating the $NO_2$ SCD data to v2.0. We will update our POMINO-GEMS algorithm accordingly, once the updated official $NO_2$ product becomes available to provide necessary inputs for our research product. In addition, in the conversion from $NO_2$ VCDs to surface concentrations, we use a constant correction factor of 2 to account for the strong $NO_2$ vertical gradient near the surface. This simple treatment does not account for the diurnal variation of the correction factor, and thus may introduce errors in the derived surface $NO_2$ concentrations."

**References:**

Kong, H., Lin, J., Zhang, Y., Li, C., Xu, C., Shen, L., Liu, X., Yang, K., Su, H., and Xu, W.: Unexpected high NOX emissions from lakes on Tibetan Plateau under rapid warming, 10.21203/rs.3.rs-1980236/v1,

2022.

---

## Author Comment (AC3)

**Authors' response to comments from Anonymous Referee #3**

**General comments:**

The Korean GEMS satellite is the first of a series of geostationary satellite instruments providing hourly observations of key air pollution species, including $NO_2$. These data are of large interest for air quality studies. As the current operational GEMS tropospheric $NO_2$ product still has some deficiencies, there is a need for improvements, and this manuscript is aiming at improving on that.

The approach taken in this study is to use the reprocessed PAL version of the TROPOMI $NO_2$ product to determine a pixel specific slant column offset of the GEMS data at TROPMI overpass time, and to apply it to all GEMS measurements. The stratospheric correction is based on TM5 stratospheric VC data, again from the TROPOMI product, together with the diurnal variation taken from a GEOS-Chem run. Cloud correction and AMFs are computed using an updated version of the POMINO retrieval framework. The algorithm is applied to three months of data and the resulting columns compared to TROPOMI data, MAX-DOAS observations and in-situ surface measurements.

The manuscript is clearly written, covers a topic of interest to the AMT readership and reports on a relevant study. However, I have some concerns that the authors need to address before the manuscript can be accepted for publication.

We thank the Referee #3 for taking time to review our paper and for providing constructive suggestions and comments for improvement. We updated our POMINO-GEMS algorithm by replacing nested GEOS-Chem v9-02 derived stratospheric $NO_2$ VCDs with NASA GEOS-CF v1 derived stratospheric $NO_2$ VCDs, and reprocessed all retrievals. Updated validation results show great improvement in $NO_2$ diurnal patterns between POMINO-GEMS and ground-based MAX-DOAS measurements. We also use mobile-car MAX-DOAS measurements in the Three Rivers' Source region on the Tibetan Plateau to validate POMINO-GEMS retrievals, and good agreement is also shown in terms of $NO_2$ diurnal variation. Responses to the major and specific comments are provided below.

**Major comments:**

My main criticism about the paper is that the approach taken (correction of GEMS SCD data using TROPOMI retrievals) is a temporary solution at best. Clearly, problems in the GEMS SCD retrievals need to be solved in the spectral fit and not using an ad-hoc correction linking it to data from another satellite instrument. also, the assumption that all the problems in GEMS data can be described by a slant column offset determined at the time of TROPOMI overpass is probably not correct, as solar zenith angle and relative azimuth angle change over the day. Therefore, the most important measurement quantity of GEMS, the diurnal variation of $NO_2$, could be affected by the applied method.

It is also important to realize, that GEMS and TROPOMI data taken at the same time of the day do not have the same scattering geometry, and thus not the same AMF. The slant columns can therefore be different, even after geometric correction. These problems of the current approach need to be discussed in the manuscript.

Thank you very much for your comments.

(1) We agree that current approach to correct GEMS total $NO_2$ SCDs is a temporary solution at best, and the systematic problems in the official GEMS SCD retrieval should be solved by improving spectral fitting. As we are planning to learn and perform DOAS method to directly retrieve $NO_2$ SCDs, $NO_2$ SCDs calculated from this temporary solution can be a valuable reference to evaluate our product in the future. We have added more discussion on this limitation.

In Line 255-257, we added:

"We also note that our simple correction is a temporary solution before the aforementioned systematic problems in the official GEMS SCD retrieval are solved by improving spectral fitting."

In Line 757-765, we added:

"To address the systematic overestimation and stripes problems in the original GEMS data, we correct GEMS total $NO_2$ SCDs by using TROPOMI data as a temporary solution. For example, we implement a simple geometric correction to combine GEMS and TROPOMI total $NO_2$ SCDs, but their differences in scattering geometry are only partly accounted for. Thus this correction works well in most regions, but may introduce SCD uncertainties up to $0.9 \times 10^{15}$ molec. cm$^{-2}$ (20% – 30%) at the edge of the northwestern GEMS FOV. Currently, the Environmental Satellite Center of South Korea is updating the $NO_2$ SCD data to v2.0. We will update our POMINO-GEMS algorithm accordingly, once the updated official $NO_2$ product becomes available to provide necessary inputs for our research product."

(2) The assumption of the correction of GEMS SCD data using TROPOMI retrievals is not clearly presented before, so we have added more discussion about the assumption in our geometric correction.

In line 237-241, we added:

"In Eq. (2), we implement a simple geometric correction (concerning SZAs and VZAs) for AMFs instead of using the actual AMFs; the latter could account for the differences in relative azimuth angles and other factors. Specific derivation of this assumption is given in Section 1 of the Supplement Information (SI). The correction is assumed to be acceptable with an extra uncertainty introduced to the total $NO_2$ SCDs, as will be further discussed in Section 3.5."

In Line 680-683, we added:

"Given the assumption we made in adjusting GEMS total SCDs to match TROPOMI values, we tentatively estimate the error in our corrected total SCD data to be $0.5 – 0.7 \times 10^{15}$ molec. cm$^{-2}$ (10% in a relative sense) for most regions and $0.9 \times 10^{15}$ molec. cm$^{-2}$ (20% – 30%) at the edge of the northwestern GEMS FOV."

(3) It's true that TROPOMI-guided correction for GEMS total $NO_2$ SCDs could affect the diurnal variations of $NO_2$ from GEMS observations, so we have added additional comparisons and discussion about the diurnal variations of uncorrected GEMS $NO_2$ VCDs. The comparison results show that no significant influence on the diurnal variation of POMINO-GEMS tropospheric $NO_2$ VCDs is brought in through TROPOMI-based correction for total $NO_2$ SCDs.

In Line 569-577, we added:

"As we use TROPOMI total $NO_2$ SCDs to correct those of GEMS, this may influence the $NO_2$ diurnal variation of original GEMS observations. Thus we also compare MAX-DOAS data with re-calculated POMINO-GEMS tropospheric $NO_2$ VCDs without correction in total SCDs (red dashed lines in Figure 9). Compared to our default POMINO-GEMS data (with correction), excluding the correction leads to lower diurnal correlation coefficients at Xuzhou, Hefei, Fudan University, Nanhui and Dianshan Lake, but higher correlation coefficients at Xianghe, Chongming and Fukue. Excluding the correction increases the NMB at three sites but decreases the NMB at five sites. We conclude that at these eight sites (in the eastern areas), no significant influence on the diurnal variation of POMINO-GEMS tropospheric $NO_2$VCDs is brought in through TROPOMI-based correction for total $NO_2$ SCDs."

In Line 586-588, we added:

"In contrast, POMINO-GEMS without total SCD correction exhibits much poorer correlation with mobile-car MAX-DOAS data, due to the erroneous increase in the afternoon."

In Line 639-640, we added:

"Note that the consistency between POMINO-GEMS and MEE data does not depend on the total SCD correction (Table S4)."

My second concern is about the comparison of GEMS and TROPOMI data shown in the manuscript. As GEMS slant columns are forced to agree with TROPOMI data, this comparison makes little sense and only shows that no technical mistake was made. The only comparisons providing additional information are those to external data.

Thank you for your comment and suggestion. The discrepancies between POMINO-GEMS and POMINO-TROPOMI v1.2.2 tropospheric $NO_2$ VCDs are caused by differences in both tropospheric $NO_2$ SCDs and AMFs. We have added detailed discussion in Section 3 of the SI. Besides, we have also added the comparison results between POMINO-GEMS and independent OMNO2 v4 and GOME-2 GDP 4.8 tropospheric $NO_2$ VCDs.

In Line 496-506, we added:

"Figure 7d-f and g-i show the comparison results of POMINO-GEMS tropospheric $NO_2$ VCDs with OMNO2 v4 on a $0.25° \times 0.25°$grid and GOME-2 GDP 4.8 on a $0.5° \times 0.5°$grid averaged over JJA 2021, respectively. POMINO-GEMS $NO_2$ VCDs exhibit good spatial consistency with the two independent products ($R$ = 0.87 and 0.83), although with slightly lower values than OMNO2 v4 (by 16.8%) and GOME-2 GDP 4.8 (by 1.5%). These VCD differences are expected, considering the differences in the retrieval algorithm. For example, the POMINO-GEMS algorithm implements explicit aerosol corrections in the radiative transfer calculation, while OMNO2 v4 and GOME-2 GDP 4.8 treat aerosols as "effective clouds". POMINO-GEMS accounts for the anisotropy of surface reflectance by adopting MODIS BRDF coefficients, whereas OMNO2 v4 and GOME-2 GDP 4.8 use geometry-dependent and regular LER, respectively. The horizontal resolution of a priori $NO_2$ profiles in POMINO-GEMS is 25 km (and interpolated to 2.5 km), $1° \times 1.25°$in OMNO2 v4 and $1.875° \times 1.875°$in GOME-2 GDP 4.8."

My third point is, that the uncertainty discussion is very superficial and in my opinion not correct. The

SC uncertainty should be driven by shot noise and therefore be described as an absolute, not a relative uncertainty. The overall uncertainty of 0.2E15 molec/cm$^2$ derived for the tropospheric SCDs appears very low, but it is anyway not clear if this is the uncertainty for an individual GEMS measurement, a monthly average, or the three monthly average discussed here. this discussion needs to be improved.

Thank you very much for your comment. We have re-written Section 3.5 to improve the discussion about the error estimates. All uncertainties discussed here are for the summertime retrieval. For the uncertainty of NO$_2$ slant columns, we have discussed it both using an absolute value and in a relative sense. The relative uncertainty of NO$_2$ SCDs is used for following estimation of relative uncertainty of tropospheric NO$_2$ VCDs. We agree that the overall uncertainty for tropospheric NO$_2$ SCDs is underestimated, and updated analysis has been added in the revised manuscript.

In the data availability section, it is stated that the data is available through http://www.pku-atmos-acm.org/acmProduct.php/. This does not appear to be the case and data could therefore not be checked for this review.

Thank you very much for your comment. At first, we were processing the retrieval data beginning in December 2020 and hadn't upload them online. Since we updated our retrieval algorithm by using NASA GEOS-CF v1 product, we now are reprocessing all the data and will upload them for public use as soon as possible.
We have changed the data availability statement to "The POMINO-GEMS NO$_2$ data will be freely available soon at the ACM group product website (http://www.pku-atmos-acm.org/acmProduct.php/)."

**Minor comments:**

Line 188: Isn't the current GEMS NO$_2$ product provided at 3.5 × 8 km$^2$?

Yes, the current GEMS NO$_2$ product is provided at 3.5 × 8 km$^2$, but the spatial resolutions of other trace gases are different. Therefore we decide to quote the statement in Kim et al. (2020) to generally describe the spatial resolution of GEMS products.

Line 128: How does the known GEMS uncertainty in irradiances affect the reflectances and thereby cloud retrievals?

The uncertainties in the measured radiances at the top of atmosphere and extraterrestrial solar irradiances can directly affect the cloud fraction retrieval, and also be propagated to the uncertainties of DOAS-fitted continuum reflectances and O$_2$-O$_2$ SCDs used for the inversion of cloud-top pressure.
Currently we don't exactly know the uncertainty in radiances measured by GEMS instrument. In our POMINO-GEMS algorithm, we re-retrieve cloud parameters in order to assure the consistency of ancillary parameters used for cloud and NO$_2$ retrieval, such as aerosol optical parameters and surface reflectance.

Line 262: this ad hoc factor needs to be mentioned again when later comparing the retrievals with the in-situ observations.

Thank you for your suggestion. We have added more discussion about the limitation of the ad hoc factor in section 3.4 in the revise manuscript.

In line 606-608, we added:

"These differences reflect errors in POMINO-GEMS $NO_2$ VCDs, in the conversion from tropospheric VCDs to surface concentrations, and in MEE data (due to potential contamination by nitric acid and organic nitrates (Liu et al., 2018))."

In Line 631-639, we added:

"The discrepancies between POMINO-GEMS and MEE surface $NO_2$ concentrations at different hours are likely caused by the assumed constant correction factor of 2 to account for the vertical gradient of $NO_2$ from the height of ground instrument to the center of the first model layer (Section 2.2). In the morning when the PBL is low, most $NO_2$ molecules are near the ground and the vertical gradient of $NO_2$ over polluted regions is the largest in the daytime, so the factor of 2 may lead to underestimation of derived surface $NO_2$ concentrations. In contrast, in the afternoon, the PBL mixing is much stronger and the vertical gradient of $NO_2$ is much smaller, thus the factor of 2 may lead to overestimated surface $NO_2$ concentrations."

Line 277: please provide a bit more information on this – how many data points were excluded? What exactly were the criteria?

Thank you for your suggestion. We have added more information about the Grubbs statistical test, and have also shown the comparison between the original data and those after excluding outliers in Figure S7.

In line 357-361, we added:

"The Grubbs statistical test, which is used to detect outliers in a univariate data set assumed to exhibit normal distribution (Grubbs, 1950), is performed to exclude outliers in both MAX-DOAS and satellite data before comparison. Only one data pair from Fudan University site is identified as an outlier and removed (Figure S7), and we get 1348 matched hourly data pairs in total."

Figure 4: What are the regions shown in grey in the figure? Are these negative values of missing data?

The regions in grey mean there are no GEMS observations or valid retrievals in June-July-August 2021. They are either because of the spatial limitation of GEMS FOV, or because the pixels are excluded due to the quality control criteria. We have added the note in the caption of all corresponding figures.

Figure 6 / Line 323: I do not find the discussion of the observed increase in $NO_2$ convincing. The observed changes are large and have clear patterns, and I suspect they are retrieval artefacts.

Thank you very much for your comment. Previous studies have discovered the $NO_x$ emissions from natural sources such as soil and lakes. After using more reasonable stratospheric $NO_2$ information from GEOS-CF v1 product, the increase in $NO_2$ over this region is still evident, so we believe it is hardly a

retirval artefact. We will further explore this issue in the future.

Figure 7: As discussed above, the only surprise with this figure is that the agreement is not even better.

Thank you for your comment. We have added detailed analysis for the differences between POMINO-GEMS and POMINO-TROPOMI v1.2.2 tropospheric $NO_2$ VCDs in Section 3 of the SI.
Besides, we have also compared POMINO-GEMS tropospheric $NO_2$ VCDs with those of external OMNO2 v4 and GOME-2 GDP 4.8 products. Comparison results have been shown in the reply to the second major comment.

Line 433: Maybe mention that the main difference between column and surface concentrations is that the column is insensitive to boundary layer height changes.

Done.

Line 443: Please provide information on for what the uncertainty calculations are made – individual measurements or averages?

Our uncertainty analysis is for the general summertime retrieval.

Line 448: why should SCD have a relative uncertainty?

Thank you for your comment. According to previous studies, the SCD uncertainty can both be described using an absolute value and in a relative sense. Here, we use the relative uncertainty of $NO_2$ SCDs to estimate the relative uncertainty of tropospheric $NO_2$ VCDs. We have updated our discussion about the SCD uncertainty in the revised manuscript.

In Line 676-683, we added:
"As described in Section 2, we calculate hourly total SCDs based on the original GEMS SCD data and daily TROPOMI-guided corrections. According to the GEMS ATBD of $NO_2$ retrieval algorithm, the SCD errors from the DOAS method are < 5.65% at high-$NO_2$ conditions ($NO_2$ VCD > $1 \times 10^{15}$ molec. cm$^{-2}$) (Lee et al., 2020). The $NO_2$ SCD errors of TROPOMI are reported to be $0.5 – 0.6 \times 10^{15}$ molec. cm$^{-2}$ (10% in a relative sense) (Van Geffen et al., 2022). Given the assumption we made in adjusting GEMS total SCDs to match TROPOMI values, we tentatively estimate the error in our corrected total SCD data to be $0.5 – 0.7 \times 10^{15}$ molec. cm$^{-2}$ (10% in a relative sense) for most regions and $0.9 \times 10^{15}$ molec. cm$^{-2}$ (20% – 30%) at the edge of the northwestern GEMS FOV."

**References:**

Grubbs, F. E.: Sample Criteria for Testing Outlying Observations, The Annals of Mathematical Statistics, 21, 27-58, 10.1214/aoms/1177729885, 1950.
Kim, J., Jeong, U., Ahn, M.-H., Kim, J. H., Park, R. J., Lee, H., Song, C. H., Choi, Y.-S., Lee, K.-H., Yoo, J.-M., Jeong, M.-J., Park, S. K., Lee, K.-M., Song, C.-K., Kim, S.-W., Kim, Y. J., Kim, S.-W., Kim, M., Go, S., Liu, X., Chance, K., Chan Miller, C., Al-Saadi, J., Veihelmann, B., Bhartia, P. K., Torres, O., Abad, G. G., Haffner, D. P., Ko, D. H., Lee, S. H., Woo, J.-H., Chong, H., Park, S. S., Nicks, D., Choi,

W. J., Moon, K.-J., Cho, A., Yoon, J., Kim, S.-K., Hong, H., Lee, K., Lee, H., Lee, S., Choi, M., Veefkind, P., Levelt, P. F., Edwards, D. P., Kang, M., Eo, M., Bak, J., Baek, K., Kwon, H.-A., Yang, J., Park, J., Han, K. M., Kim, B.-R., Shin, H.-W., Choi, H., Lee, E., Chong, J., Cha, Y., Koo, J.-H., Irie, H., Hayashida, S., Kasai, Y., Kanaya, Y., Liu, C., Lin, J., Crawford, J. H., Carmichael, G. R., Newchurch, M. J., Lefer, B. L., Herman, J. R., Swap, R. J., Lau, A. K. H., Kurosu, T. P., Jaross, G., Ahlers, B., Dobber, M., Mcelroy, C. T., and Choi, Y.: New Era of Air Quality Monitoring from Space: Geostationary Environment Monitoring Spectrometer (GEMS), Bulletin of the American Meteorological Society, 101, E1-E22, 10.1175/bams-d-18-0013.1, 2020.

Lee, H., Park, J., and Hong, H.: Geostationary Environment Monitoring Spectrometer (GEMS) Algorithm Theoretical Basis Document NO2 Retrieval Algorithm, 2020.

Liu, M., Lin, J., Wang, Y., Sun, Y., Zheng, B., Shao, J., Chen, L., Zheng, Y., Chen, J., Fu, T. M., Yan, Y., Zhang, Q., and Wu, Z.: Spatiotemporal variability of NO2 and PM2.5 over Eastern China: observational and model analyses with a novel statistical method, Atmos. Chem. Phys., 18, 12933-12952, 10.5194/acp-18-12933-2018, 2018.

Van Geffen, J., Eskes, H., Boersma, K. F., and Veefkind, P.: TROPOMI ATBD of the total and tropospheric NO2 data products, 2022.

---

## Author Response (AR2)

**Authors' response to comments from Anonymous Referee #1**

**General comments:**

I have gone over the changes made in response to my previous comments as Reviewer #1, and am satisfied that all my comments have been addressed.

Thank you very much for taking time to review our revised paper.

**Specific comments:**

This reference for OMNO2 v4 would be better:
https://amt.copernicus.org/articles/14/455/2021/amt-14-455-2021-discussion.html

Thank you for your suggestion. We have updated the reference in the revised manuscript.

**Technical comments:**

Line 64: Change "it threats" to "it threatens"

Done.

**Authors' response to comments from Anonymous Referee #3**

**General comments:**

The authors have put a lot of work in this revision, and have improved on the data, the validation and the presentation. I therefore recommend the manuscript for publication.

Thank you very much for taking time to review our revised paper.

I have however some concerns, and they are about Figure 9.

Comparing the old and the new version of the data, it is surprising to see that even at polluted places such as Xianghe, the new GEMS data differs by up to a factor of two from the previous version. As far as I could see, the only difference in the two versions is the treatment of the diurnal variation of the stratosphere. I'm surprised to see that the stratospheric correction has such a large impact and think that this hints at a larger contribution of this term to the overall uncertainty.

Thank you very much for your comment. In the previous version, the very little diurnal variation of stratospheric $NO_2$ is actually incorrect because of the insufficient stratospheric chemistry in nested GEOS-Chem v9-02 simulations. After using the stratospheric $NO_2$ from GEOS-CF v1 dataset, the

retrieval results are much better and also reasonable. Besides, GEOS-CF v1 stratospheric $NO_2$ spatiotemporally correlates very well with that in the TROPOMI-PAL v2.3.1 product, and also shows similar diurnal variation characteristics compared with previous studies. Therefore, the large difference is caused by the wrong stratospheric $NO_2$ correction in the previous version.

A second concern I have about this figure is that also the MAX-DOAS values have changed, at least for Xianghe and Dianshan Lake, possibly also for other locations. Please explain why that's the case.

I'm quite sorry that the labels in the original manuscript are mismatched to the corresponding subfigures. In the original manuscript, Figure 9e should be for Xianghe, Figure 9f for Dianshan Lake, and Figure 9g for Chongming, respectively. The MAX-DOAS values are the same as before.

I'm also confused why things have improved so much in Figure 9 with the new data version, while in Figure 8a, which is based on the same data, no improvement is apparent. Please explain.

Thank you for your comment. As shown in Figure 9, the update of stratospheric $NO_2$ data results in a great improvement in terms of $NO_2$ diurnal variation, but the normalized mean biases are still site-dependent and don't change very much. Compared with updated POMINO-GEMS tropospheric $NO_2$ VCDs, MAX-DOAS $NO_2$ VCDs are still higher at the Xuzhou, Fudan University, Nanhui and Chongming sites, but lower at the Hefei, Xianghe, Dianshan Lake and Fukue sites. Therefore, the improvement of spatiotemporal correlation and normalized mean bias is not very apparent, but the correlation of $NO_2$ diurnal variation becomes much better.